Potential transoceanic dispersal of Geodia cf. papyracea and six new tetractinellid sponge species descriptions within the Hawaiian reef cryptofauna

Nunley Rachel M.
http://orcid.org/0009-0008-9526-1241 Rutkowski Emily C.
http://orcid.org/0000-0001-6339-4340 Toonen Robert J.
http://orcid.org/0000-0003-4741-0261 Vicente Jan vicentejan@gmail.com
Hawai‘i Insitute of Marine Biology, University of Hawai‘i at Mānoa , Kāne‘ohe Bay, Hawai‘i , United States
Guy-Haim Tamar
Electronic publication date: 2025 Feb 17
Publication date: 2025
Volume: 13
Electronic Location ID: e18903
Received 2024 Sep 6; Accepted 2025 Jan 6
Copyright: © 2025 Nunley et al.
Copyright year: 2025
Copyright holder: Nunley et al.
License: This is an open access article distributed under the terms of the Creative Commons Attribution License, which permits unrestricted use, distribution, reproduction and adaptation in any medium and for any purpose provided that it is properly attributed. For attribution, the original author(s), title, publication source (PeerJ) and either DOI or URL of the article must be cited.
License URL: https://creativecommons.org/licenses/by/4.0/

Keywords: Porifera, Astrophorina, Phylogeny, Systematics, Cryptic fauna, Alien species

Funding: NSF Postdoctoral Research Fellowship in Biology #1612307 NSF awards OA#1416889 & OCE# 2048457 National Oceanic and Atmospheric Administration’s Ocean Acidification Program #60046834 Funding was provided by an NSF Postdoctoral Research Fellowship in Biology (#1612307), NSF awards OA#1416889 & OCE# 2048457, and by the National Oceanic and Atmospheric Administration’s Ocean Acidification Program (#60046834). The funders had no role in study design, data collection and analysis, decision to publish, or preparation of the manuscript.

==============================
Kāne‘ohe Bay has historically been known for the introduction of alien species from the Caribbean and the Western Indo-Pacific. Recent efforts that explore the reef cryptofauna have shown that in addition to the diversity of non-indigenous species, patch reef environments are rich with undescribed species. Here we integrate molecular phylogeny and systematics to distinguish introduced species from those that are potentially native or endemic. We focus on the order Tetractinellida and document the potential transoceanic dispersal of Geodia papyracea from the Caribbean to Hawai‘i. Our integrative approach allowed us to describe new species of Stelletta (Stelletta kela sp. nov., Stelletta hokunalohia sp. nov., Stelletta kuhapa sp. nov., Stelletta hokuwanawana sp. nov., Stelletta apapaola sp. nov.) and one new species of Stryphnus (Stryphnus huna sp. nov.); all collected from the reef cryptofauna via the use of Autonomous Reef Monitoring Structures. Specimens were barcoded using 28S and COI molecular markers, providing insights into the phenotypic plasticity of sponges and the phylogenetic placement of these new species based on morphological characters. Using both molecular phylogeny and traditional taxonomy enhances the accuracy of species identification and classification, contributing to a broader understanding of sponge biodiversity within the Hawaiian archipelago.

Introduction

Global biodiversity surveys which incorporate detailed taxonomic assessments of species are crucial for accurately determining the geographic origin of species and helps unveal the evolutionary history of those species in an ecosystem (Quintero et al., 2015; Sandall et al., 2023). This is of particular relevance to the natural history of the Hawaiian island archipelago, which ranks among the highest in the world for both the number of marine alien invasive species and the success of these species in outcompeting and displacing native marine fauna (Conklin & Smith, 2005; Coles et al., 2007; Carlton & Eldredge, 2009). Many alien marine species, including sponges, are difficult to identify without the use of thorough taxonomic evaluation beyond the unaided eye (Hooper & Van Soest, 2002). The high number of alien species, combined with the lack of a proper taxonomic database for sponges throughout Oceania has limited our ability to classify newly discovered sponge species, often leaving them labeled as cryptogenic (neither clearly native nor introduced) (Carlton, 1996; Conklin & Smith, 2005). Constraint in this regard has caused a knowledge gap in our ability to understand how sponges from rich cryptic coral reef communities have evolved throughout the Pacific.

Cryptic reef habitats are defined as low-light environments created by overhangs, caves, or spaces between branching coral colonies (Choi, 1982) providing protected recruitment habitats for sciophilous, sessile invertebrates. Sponges make up a majority of the species that reside in these habitats (Kornder et al., 2021) and absorb dissolved organic matter (DOM) that provides energy and nutrients to coral reefs (de Goeij & van Duyl, 2007). Sponges are responsible for the uptake of DOM and help assimilate these nutrients by converting DOM into particulate organic matter (POM) (de Goeij et al., 2013) which can feed detritivores and influence reef trophodynamics. Cryptic sponges are integral to the nutrient dynamics of coral reefs yet remain understudied due to the difficulties in accessing these habitats, and long-standing biases in research priorities (Caldwell et al., 2024; Vicente et al., 2022b).

The vast majority of sponges reported in Hawai‘i have generally come from bays and harbors where they can grow to be conspicuous (Carlton & Eldredge, 2009; Coles et al., 1997; Coles, DeFelice & Eldredge, 2002). However, a diversity of inconspicuous sponges living in cryptic spaces, and therefore difficult to access, have been largely overlooked by previous studies (Vicente et al., 2022b). Historically, the isolation of most Pacific islands has impeded detailed sponge surveys from being conducted in relation to better-studied geographic regions such as the Caribbean and the northeast Atlantic (van Soest et al., 2012). The Hawaiian Islands are the most isolated archipelago on the planet, with islands that vary in geologic age (Fletcher et al., 2008) as well as in reef type, ranging from those that are majority rock substrate (lava beds or boulders) to others with mostly calcareous and sand substrate (Grigg, 1983; Jokiel et al., 2004). The extreme isolation of the archipelago combined with the wide diversity of cryptic reef habitats that remain largely unexplored implies a potentially undiscovered endemism among the Hawaiian tropical reef sponges.

One reason for this knowledge gap is the difficulty of exploring the cryptic reef environments which are typically embedded deep within the three-dimensional matrix of a thriving coral reef. It is nearly impossible to access these spaces without destroying the surrounding reef, but Autonomous Reef Monitoring Structures (ARMS) (Brainard et al., 2009; Knowlton et al., 2010; Zimmerman & Martin, 2004) have proved effective at providing a standardized method for sampling these habitats. By offering an artificial facsimile of this habitat, species such as the cryptic sponge fauna recruit naturally to these structures which can be removed for study without destruction of the reef itself (Brainard et al., 2009; Knowlton et al., 2010; Zimmerman & Martin, 2004). Further, ARMS have already been successfully used to survey and monitor a wide variety of sponge species in Kāne‘ohe Bay (Vicente et al., 2022a, 2022b), including tetractinellid sponges.

Tetractinellida is the second most speciose order (~1,183) within the class Demospongiae (de Voogd et al., 2024), but these sponges present taxonomic challenges due to their morphological variability. For example, Astrophorina, a suborder of Tetractinellida, has one of the most diverse spicule compositions within demosponges (Cárdenas et al., 2011). The diversity of megascleres and microscleres can be shared across the different families within the suborder, which makes it difficult to identify unique synapomorphic characters within species belonging to different families (Cárdenas et al., 2011). For example, within the family Ancorinidae, Stelletta is defined by having triaenes (sometimes absent) and some form of euaster (Uriz, 2002) and contains the most species (150) of the 16 genera in the Ancorinidae family (de Voogd et al., 2024). Additionally, Stryphnus contains 16 species and is defined by having large oxeas, triaenes, euasters, amphiasters and sanidasters (Uriz, 2002). However, some species that lack triaenes are placed within Stryphnus due to low DNA sequence divergences (Cárdenas et al., 2011). This highlights the difficulty in identifying new species based on genera definitions and morphology alone, especially when molecular data do not support these taxonomic differences. The polyphyly of Stelletta (Cárdenas et al., 2011) reveals the importance of integrating classical systematics with molecular phylogeny to accurately classify species.

Currently, there are twelve species of tetractinellids known from Hawai‘i, belonging to the genera Stelletta (S. debilis (Thiele, 1900)), Geodia (G. gibberella de Laubenfels, 1951), Erylus (E. caliculatus von Lendenfeld, 1910, E. proximus Dendy, 1916, E. rotundus von Lendenfeld, 1910, E. rotundus var. cidaris von Lendenfeld, 1910, E. rotundus var. megarhabdus von Lendenfeld, 1910, E. rotundus var. typicus von Lendenfeld, 1910, E. sollasi von Lendenfeld, 1910), Jaspis (J. digonoxea (de Laubenfels, 1950), J. pleopora (de Laubenfels, 1957)), and Asteropus (A. kaena (de Laubenfels, 1957)). Stelletta debilis, described by de Laubenfels (1951) shows morphological differences from its holotype described by Thiele (1900) in Indonesia, suggesting the Hawaiian specimen may not be a conspecific of this species, requiring further comparative analyses for accurate classification.

Tetractinellid diversity is typically concentrated in cryptic spaces (e.g., coral crevices, under rocks, caves) or deep-sea habitats with minimal sun exposure (Cárdenas et al., 2009; Díaz et al., 2024) making ARMS a great tool for uncovering these species. Before ARMS deployment in Hawai‘i, only one tetractinellid species (Jaspis digonoxea) was known to occur in Kāne‘ohe Bay, and eleven from throughout the archipelago. However, using 35 modified ARMS and six full ARMS (Vicente et al., 2022b; Fig. S1) uncovered seven tetractinellid species at a single site, each showing >1% genetic variation in COI and 28S rRNA sequences (Vicente et al., 2022b). This research is essential for uncovering tetractinellid species abundance throughout the Hawaiian Islands for a better understanding of their ecosystem functions in cryptic reef environments.

Due to the difficulties in identifying tetractinellid sponges, we use an integrative taxonomic approach to determine whether species from ARMS match those from historical collections in Hawai‘i. We use a multi-locus (28S rRNA and COI barcoding) approach since previous barcoding and metabarcoding efforts using COI universal primers failed to distinguish some putative Tetractinellida species (Vargas et al., 2012; Timmers et al., 2020; Vicente et al., 2022b). This study contributes to the richness and biodiversity of Hawaiian reefs and creates a database for tetractinellid species to aid in identifying species and help determine the geographic origin within this unique group in future studies. By integrating practices in systematics and molecular phylogenetics, we describe five new Stelletta species, one new Stryphnus species, and provide documentation of a possible introduction from the Caribbean (Geodia cf. papyracea) in Hawai‘i.

Materials and Methods

Sponge collection

Sponges were photographed in situ and collected from ARMS inside mesocosms at the Hawai‘i Institute of Marine Biology (HIMB) on Moku o Lo‘e (Coconut Island) (21.4335° N, −157.78634°W), O‘ahu, from ARMS that had been soaking for 6 years deployed on a natural reef environment adjacent to Moku o Lo‘e, and from the surface of outflow pipes on Moku o Lo‘e (Coconut Island), O‘ahu. Sampling occurred bimonthly between 2016–2019. Field observations and measurements of morphology, color, consistency, surface, and oscules for each specimen were recorded. Samples were preserved in 95% ethanol and when enough material was available, were also fixed in 4% paraformaldehyde (PFA) for 24 h and then transferred to 70% ethanol. Some specimens were subsampled and deposited in the Florida Museum of Natural History (catalog number beginning with acronym UF) in Florida, USA, and the Bernice Pauahi Bishop Museum (catalog number beginning with acronym BPBM) in O‘ahu, USA. Specimens without a UF catalog number, along with all spicule preparations and section slides were only stored at the Bernice Pauahi Bishop Museum. Samples from Kāne‘ohe Bay were collected under special activities collection permits SAP2018–03 and SAP2019–06 (covering the period of January 13, 2017, through April 10, 2019) as well as HIMB collection permits SAP2022-22 and SAP2023-31. Samples from 2016 were collected from mesocosms where no permit was required.

DNA extraction, sequencing, and assembly

Subsamples of sponge tissue (30 mg) were removed from each specimen and were preserved in 95% ethanol and processed for DNA extractions. Methods for DNA extractions, polymerase chain reactions (PCR), and Sanger sequencing are found in Vicente et al. (2022b) with minor modifications as follows: 1. We used the 28S 63MODF (5′-ACC CGC TGA AYT TAA GCA TAT HAN TMA-3′) forward with the 1072RV (5′-GCT ATC CTG AGG GAA ACT TCG G-3′) (Medina et al., 2001) reverse primer, or the 28S C1′ ASTR FWD (5′-ACC CGC TGA ACT TAA GCA T-3′) (Cárdenas et al., 2010) combined with the 28S 1072RV reverse primer to amplify the 28S rRNA. 2. We used the COI dgLCO1490 (5′-GGT CAA CAA ATC ATA AAG AYA TYG G-3′) and COI dgHCO2198 (5′-TAA ACT TCA GGG TGA CCA AAR AAY CA-3′) (Meyer, Geller & Paulay, 2005) to amplify the Folmer regions of the COI fragment. The PCR program consisted of an initial denaturation at 95 °C for 3 min, followed by 34 cycles of denaturation for 30 s at 95 °C, annealing at 45 °C for 45 s, and extension for 1 min at 72 °C. A final extension hold at 72 °C for 5 min finished the reaction. Forward and reverse reads from the Sanger sequences were assembled, trimmed, and edited by eye using Geneious R6 (Kearse et al., 2012). Sequences were checked for contamination using the BLAST (Altschul et al., 1990) function in GenBank and results that showed >85% sequence identity to a sponge were used for the alignment.

Phylogenetic analysis

The Geneious alignment function (Geneious R10) with default parameters (i.e., nucleotide global alignment with free end gaps, 65% similarity cost matrix, gap open penalty of 12, gap extension penalty of 3, and two refinement iterations) was used for aligning 28S rRNA and COI sequences. The 28S alignment consisted of 213 bp and the concatenated alignment consisted of 676 bp of the 28S and 589 bp for the COI which resulted in 1,196 bp of the 28S+COI gene. RaxML (Stamatakis, 2006) included in Geneious was used for maximum likelihood (ML) analysis with the GTR+GAMMA model of nucleotide substations, 100 starting maximum parsimony trees, and 2,000 bootstrap replicates. Phylogenetic trees were rooted on Cinachyrella apion (Uliczka, 1929), HM592753.1 and HM592667.1 for 28S and COI phylogenies respectively. All accession numbers pertaining to each new species and each Geodia cf. papyracea replicate are available in Table S1. For 28S and COI alignments, see text files in the Supplemental Material.

Morphological analysis

Sponge pieces containing both ectosomal and choanosomal tissue fixed in either 4% PFA or 95% ethanol were transferred to 70% ethanol. Sponge pieces were dehydrated in alcohol using a series of 35%, 50%, 70%, and 100% before being embedded in paraffin using a Leica EG1150H paraffin embedding station and a Leica EG1150C cold plate to harden the paraffin. The paraffin sections (100–300 μm thick) were then cut perpendicular to the surface of the sponge through the ectosome and choanosome using a Microm HM 340E Microtome. These sections were placed on glass slides and rinsed with xylene to remove the paraffin. Permount was added to the sections, covered with a cover glass, and then left to dry before imaging under light microscopy. Small pieces were also boiled in 67–69% pure nitric acid for 1–2 min or until the solution turned clear and all organic matter dissolved. Spicules were left to settle (~15 min), and the acid supernatant was discarded. Spicules were then suspended with distilled water and decanted three times to remove the acid, before adding 95% ethanol for long-term storage. Spicules were suspended by shaking, and a few drops from the solution were observed under light microscopy, photographed, and measured using ImageJ (Abràmoff, Magalhães & Ram, 2005) relative to a stage micrometer. A minimum of 30 megascleres and 10–15 microscleres were measured per species (unless noted otherwise) for lengths and widths (expressed as minimum–mean–maximum, length × width in μm). A few drops of the spicule suspension were added to a stub, air-dried, coated in gold, and imaged under a Hitachi S-4800 FESEM Scanning Electron Microscope (SEM) at the Biological Electron Microscope Facility at the University of Hawai‘i at Mānoa. Spicules from each paratype of every species were analyzed to confirm similarities in composition and size.

Literature from previous species descriptions for comparative analyses

All previously described Tetractinellida species in Hawai‘i are included for comparison in Table S2. Sponges that shared morphological characters (i.e., spicule composition) in the genera Stelletta, Asteropus, Jaspis, and Stryphnus were included in Table S3 which lists species closely related to the new species described here. Due to the high risk of invasive species in Hawai‘i, sponges from around the world were analyzed for comparative purposes. Species locations were described following the Marine Ecoregions in Box 1 of Spalding et al. (2007). Species from cold climates (e.g., Arctic, Adriatic Sea, etc.) were not used as comparative literature.

The electronic version of this article in Portable Document Format (PDF) will represent a published work according to the International Commission on Zoological Nomenclature (ICZN), and hence the new names contained in the electronic version are effectively published under that Code from the electronic edition alone. This published work and the nomenclatural acts it contains have been registered in ZooBank, the online registration system for the ICZN. The ZooBank LSIDs (Life Science Identifiers) can be resolved and the associated information viewed through any standard web browser by appending the LSID to the prefix http://zoobank.org/. The LSID for this publication is: (urn:lsid:zoobank.org:pub:F858D9D7-986F-4E56-9C27-B20AC2C12D81). The online version of this work is archived and available from the following digital repositories: PeerJ, PubMed Central SCIE and CLOCKSS.

Results

Systematic Descriptions:

Phylum Porifera Grant, 1836

Class Demospongiae Sollas, 1885

Subclass Heteroscleromorpha Cárdenas, Pérez & Boury-Esnault, 2012

Order Tetractinellida Marshall, 1876

Suborder Astrophorina Sollas, 1887

Family Ancorinidae Schmidt, 1870

Genus Stelletta Schmidt, 1862

Stelletta kela sp. nov. (Figs. 1, 2, Table 1) urn:lsid:zoobank.org:act:D6825A92-3012-4C8C-AAE5-C510FBA1BF73.

Figure 1 In situ photos of Stelletta kela sp. nov.

(A) Holotype (UF 3970/BPBM C1644); (B) paratype (BPBM C1648); (C) paratype (BPBM C1635); (D) paratype (BPBM C1643); (E) paratype (UF 3971/BPBM C1650); (F) paratype (BPBM C1649). Protruding spicules are indicated by white arrows in (A) and (E).

Figure 2 Skeleton and spicule composition of Stelletta kela sp. nov.

(A) Cross section of paratype (UF 3971/BPBM C1650); (B) cross section of paratype (BPBM C1635) with a black arrow point to subcortical cavities (sc); (C) zoomed in cross section of paratype (UF 3971/BPBM C1650); (D–N) SEM of spicules from holotype (UF 3970/BPBM C1644); (D) anatriaene; (E) orthotriaene; (F, G,) oxea; (H) plagiotriaene; (I) zoomed in plagiotriaene cladome; (J) zoomed in anatriaene cladome; (K) zoomed in orthotriaene cladome; (L–N) acanthostrongylasters.

Table 1 Spicule measurements for Stelletta kela sp. nov.

Voucher	Anatriaenes	Orthotriaenes	Plagiotriaenes	Oxeas	Microscleres	
	Rhabdome	Cladome	Rhabdome	Cladome	Rhabdome	Cladome		Acanthostrongylasters	
	Length × Width	Diameter	Length × Width	Diameter	Length × Width	Diameter	Length × Width	Diameter	
UF 3970/BPBM C1644 (Holotype)	688–1,058.5–1,403.7 × 10.6–14.1–18	76.6–89.4–108.5	486.8–920.7–1,286.3 × 16.9–26.2–34.8	151.2–209.2–279	202–353.8–643.6 × 9.4–14.4–22.3	55–97.6–158.7	638.9–847.6–984 × 10.1–17–23	6.9–9.7–10.7	
UF 3971/BPBM C1650	770.8–1,092.9–1,722.9 (n = 26) × 7.5–14.6–21.7	73.5–10.6–133.2	677.3–915.1–1,181.6 × 20.8–31.9–37	170.9–242.3–291.7	185.1–385.3–694.7 (n = 16) × 8.2–15.1–25.3 (n = 16)	60–116.7–184.5	383.7–842.6–1,174.5 × 5.8–18.7–26.1	9.1–10.3–11.5	
BPBM C1635	752.9–978.3–1,116.1 (n = 7) × 12.5–16.8–23	67.9–137.1–109.8 (n = 19)	617.4–866.5–1,187.8 (n = 9) × 23.1–29.1–37.3 (n = 15)	212.5–259.8–288 (n = 9)	290.4–449.7–650.1 (n = 11) × 14.2–21.9–28.4 (n = 12)	88.1–134.6–189.9 (n = 9)	792.8 × 20.9 (n = 1)	7.6–9.2–11.2	
BPBM C1648	501.1–871.6–1,180.8 × 9.7–12.5–15.8	46.9–67.9–89.1	451.9–667.3–775.2 × 20.7–31–40.2	161.9–196.9–217.6	112.3–266.4–497.1 (n = 18) × 6.2–12.7–21.3 (n = 27)	44.2–90.7–178 (n = 20)	510.7–688.4–843.5 × 7–13–20.1	–	
BPBM C1649	364.2–461.7–538.1 × 8.4–9.7–11.5	32.2–39.9–44.9	518.6–607.4–684.6 × 19.3–23.9–29.6	128.8–153.2–180.4	209.8–323.9–415.1 (n = 12) × 9.7–12.4–15.4 (n = 12)	44.6–71.9–110.9 (n = 12)	448.8–630.7–776.1 × 7.6–12.2–16.3	–	
BPBM C1643	801.9–1,091.2–1,396.6 × 11.4–16.3–21.2	70.7–99.7–118.4	692.9–918.4–1,188.1 × 18.8–25.4–33.2	165.9–223.0–280.4	No length, all broken/uncommon × 9.9–20.8–26.3 (n = 6)	123.4–157.2–197.2 (n = 5)	800.4–953.4–1,063.6 × 14.9–16.3–18.1	–	
Note:

Measurements shown as minimum—average—maximum. n = 30 for megascleres and n = 10 for microscleres unless otherwise noted. Microscleres were not measured for all vouchers because microscleres were rare and only visible in cross sections. Cross sections were made for three vouchers. All measurements are presented in μm.

Ancorinidae sp. 1 and Ancorinidae sp. 2 in Vicente et al. (2022a, 2022b)

Type locality: Holotype: UF 3970/BPBM C1644, collected from ARMS in mesocosms at the Hawai‘i Institute of Marine Biology (HIMB) in Moku o Lo‘e (Coconut Island), Kāne‘ohe Bay, O‘ahu, (21.4335°N, −157.7864°W), 0.3 m, collected on 2017-06-07, coll. Jan Vicente; Paratypes: BPBM C1649 collected on 2016-12-19; BPBM C1643 collected on 2017-04-10; UF 3971/BPBM C1650 collected on 2017-09-27; BPBM C1645, BPBM C1647, BPBM C1648 collected on 2018-03-16; BPBM C1635, BPBM C1646 collected on 2018-06-11. Location, depth, and collector for all paratypes are the same as the holotype.

Diagnosis: Stelletta kela sp. nov. is distinguished by its intraspecific color variation, ranging from white, beige, brown, burgundy, and dark grey. Additionally, its spicule composition (orthotriaenes, plagiotriaenes, anatriaenes, oxeas, and acanthostrongylasters) and spicule sizes are unique characters of this species.

Description (Fig. 1): Globular, sub-globular (1–2 cm diameter) to irregular-shaped lobes (1.8 × 2.0 cm). Surface is regular and hispid from megascleres that occasionally protrude from the sponge surface (Figs. 1A, 1E). Consistency is tough and difficult to cut. A single osculum (1 mm in diameter) is present for each individual and is slightly elevated by a membrane that may contract upon contact. The color of the cortex from live specimens varies between white, beige, dark burgundy, dark grey, and different shades of brown. The color of the choanosome is consistently cream. In some cases, a mixture of these different colors can be observed on a single individual. The membrane surrounding the oscula has a distinctive blotchy color pattern of yellow, brown, and beige. In ethanol, the choanosome and cortex are white or slightly tan.

Skeleton (Figs. 2A–2C): The cortex (504–528–589 μm in length) is formed by clusters of radially organized orthotriaenes of various sizes as well as anatriaenes and plagiotriaenes although these are less abundant. Subcortical cavities (76–157–302 μm in diameter) are abundant and spaced between orthotriaene and anatriaene bundles (Fig. 2B). The cortical skeleton is primarily composed of densely packed, orthotriaenes creating a distinctive layer from the cortex. Anatriaenes and plagiotriaenes also support the choanosomal layer although these are less abundant. Oxeas are scattered throughout the choanosome and are oriented radially to orthotriaenes, plagiotriaenes, and anatriaenes. Acanthostrongylasters are scarcely distributed throughout the choanosome.

Spicules (Figs. 2D–2N, Table 1): Orthotriaenes (452–816–1,286 × 17–28–40 μm) have a smooth rhabdome with pointed tips (Fig. 2E); cladome (129–214–292 μm in diameter) is smooth with pointed tips (Fig. 2K). Plagiotriaenes (112–356–695 × 6–16–28 μm) have a smooth rhabdome with pointed tips (Fig. 2H); cladome (44–112–197 μm in diameter) is smooth with pointed tips (Fig. 2I). Although similar in shape to orthotriaenes, plagiotriaenes were much smaller in size, with clads angled at a distinct 45° angle. Anatriaenes (364–926–1,723 × 7–14–23 μm) have smooth, slender, rhabdomes with pointed tips (Fig. 2D). Some rhabdomes were slightly curved but uncommon, occasional thorns were seen on rhabdomes but these were rare. Cladomes (32–91–133 μm in diameter) are smooth with pointed tips (Fig. 2J). Some cladomes showed a slight curve on one of the clads, but were uncommon. Oxeas (383–793–1,175 × 5–16–26 μm) are mainly smooth and fusiform with pointed tips, but can rarely be blunt or asymmetrically curved (Figs. 2F, 2G). Acanthostrongylasters, rare, approximately 10 rays, with some slender arms, and others that are thicker. All arms are spiked more on tips with a mostly smooth centrum measuring 6–10–12 μm in diameter (Figs. 2L–2N).

Habitat and ecology: Specimens were collected from ARMS inside mesocosms supplied with unfiltered flow through seawater at the Hawai‘i Institute of Marine Biology (HIMB) in Moku o Lo‘e (Coconut Island). Notably, specimens were absent from ARMS on a reef surrounded by a climax sponge community throughout a 2-year monitoring period (Fig. S3 in Vicente et al., 2022a). This suggests that these species are pioneering during community development because they only appeared in habitats completely removed from a fully developed coral reef ecosystem. This species seems to be strictly sciophilous and has yet to be found on natural calcified coral reef surfaces.

Distribution: Moku o Lo‘e, Kāne‘ohe Bay, O‘ahu, Hawai‘i, 0.3 m depth.

Etymology: We use the Hawaiian word “kela” which means to project or jut out. This name was chosen to reflect the megascleres that visibly protrude from its surface.

Taxonomic remarks: The presence of orthotriaenes, plagiotriaenes, anatriaenes, oxeas, and euasters places this sponge within the family Ancorinidae which is defined as having long-rhabdome triaenes (possibly absent), oxeas, and microscleres consisting of euasters, sanidasters, or microrhabds (Uriz, 2002). While a couple of genera (Ancorina and Stelletta) within the family Ancorinidae share similarities to Stelletta kela sp. nov., members of the genus Stelletta resemble the new species the most by having abundant triaenes, oxeas, and euasters either in the choanosome or sparsely distributed throughout the sponge body. From 150 Stelletta species worldwide, 18 had morphological characteristics that were similar to Stelletta kela sp. nov. (Table S3). Matching morphological characters included the presence of orthotriaenes, plagiotriaenes, anatriaenes, oxeas, and euasters.

Of the 18 species, Stelletta durissima Bergquist, 1965, Stelletta fibrosa (Schmidt, 1870), Stelletta globulariformis (Wilson, 1902), and Stelletta purpurea Ridley, 1884 have spicule sizes within the limits of Stelletta kela sp. nov. (Table S3). However, there are some notable differences between Stelletta kela sp. nov. and each of these species. For example, Stelletta durissima has larger plagiotriaenes (rhabdomes, 500–1,400 μm; widths, 19–70 μm; cladomes, 60–487 μm), larger anatriaene cladomes (125–150 μm), wider oxeas (2–50 μm), and tylasters which are absent in Stelletta kela sp. nov. Stelletta fibrosa also lacks orthotriaenes, has larger plagiotriaene rhabdomes (500–1,200 μm), rare anatriaenes which are very abundant in Stelletta kela sp. nov., smaller anatriaene cladomes (46–80 μm), slightly larger spheroxyasters (12–18 μm), and the presence of tylasters which are absent in Stelletta kela sp. nov. Stelletta globulariformis also lacks orthotriaenes, has larger plagiotriaene rhabdomes (1,000 μm), rare anatriaenes which are abundant in Stelletta kela sp. nov., larger and thinner oxeas (1,400 × 27 μm), and tylote strongylasters which are absent in Stelletta kela sp. nov. Stelletta purpurea, has larger and thicker oxeas (1,500–2,000 × 37 μm) and tylasters which are absent in Stelletta kela sp. nov.

In Hawai‘i, there are twelve recognized tetractinellid species (Table S2), but only one congener, Stelletta debilis. The Hawaiian specimen of Stelletta debilis, sensu de Laubenfels (1951) also has notable differences to the new species. Stelletta debilis (sensu: de Laubenfels, 1951) lacks orthotriaenes and has smaller plagiotriaene cladomes (32 μm) than Stelletta kela sp. nov.

Stelletta hokunalohia sp. nov. (Figs. 3, 4, Table 2) urn:lsid:zoobank.org:act:4A8A8BBC-12FE-41DB-8543-FD1CD6D35AFC.

Figure 3 In situ photos of Stelletta hokunalohia sp. nov.

(A) Holotype (BPBM C1599); (B) paratype (BPBM C1617); (C) paratype (UF 3974/BPBM C1614); (D) paratype (BPBM C1604); (E) paratype (BPBM C1603); (F) paratype (BPBM C1621) torn specimen upon the disassembly of the ARMS showing sponge interior.

Figure 4 Skeleton and spicule composition of Stelletta hokunalohia sp. nov.

(A) Cross section of holotype (BPBM C1599); (B) cross section of paratype (UF 3973/BPBM C1601); (C) zoomed in cross section of holotype; (D–J) SEM of spicules from holotype; (D) zoomed in plagiotriaene cladome; (E) zoomed in anatriaene cladome; (F) zoomed in deformed megasclere; (G) plagiotriaene; (H) anatriaene; (I) oxea; (J) deformed megasclere. Black arrows point to spicule bouquets (bq) and subcortical cavities (sc) in A–C.

Table 2 Spicule measurements for Stelletta hokunalohia sp. nov.

Voucher	Anatriaenes	Plagiotriaenes	Oxeas	
	Rhabdome	Cladome	Rhabdome	Cladome		
	Length × Width	Diameter	Length × Width	Diameter	Length × Width	
BPBM C1599 (Holotype)	354.1–485.9–568.7 × 6.8–8.7–11.4	31.8–37.6–44.3	409.4–570.8–680.9 × 15.1–20.4–28.6	86.9–136.5–172	482.5–698.9–861.3 × 8.2–14.3–18.1	
BPBM C1603	360.1–450.5–510.9 × 6.3–8.5–10.8	33.3–39.1–44.3	314.8–551.5–696.2 × 11.2–20–24.7	82–133.7–160.4	564.9–707.8–804.4 × 9.8–13.9–17.2	
BPBM C1604	332.4–469.5–636.2 × 7.1–8.9–10.8	29.8–40–47	189.7–428.2–646.6 × 6.3–14.9–21.52	30.9–105.7–177.8	455.7–644.7–781.5 × 6.6–12.6–15.8	
BPBM C1617	294–410.6–487.6 × 5.2–7.8–10.4	28.4–36.2–45.2	337.3–462.9–599 × 10.8–17.4–23.7	58.9–112–151.4	428.3–561.2–690.3 × 10.1–12.8–15.9	
UF 3974/BPBM C1614	353.6–461.3–551.9 × 7.5–9.6–11.7	32.7–40.8–53.2	293.6–538.3–653.8 × 11.7–21.5–30.9	60.1–133.9–177	463.4–651.1–821.6 × 7–12–15.2	
BPBM C1602	336.3–389.6–451.2 × 6.4–8.6–11 (n = 27, n = 28)	28.5–32.4–36.3 (n = 22)	370.8–503.1–598.2 × 10.1–19.1–23.6 (n = 18, n = 27)	90.7–124.3–172.4 (n = 13)	421.4–553.8–674.3 × 7.6–11.6–16.5	
Note:

Measurements shown as minimum—average—maximum. n = 30 unless otherwise noted. All measurements are presented in μm.

Ancorinidae sp. 3 and Ancorinidae sp. 4 in Vicente et al. (2022a, 2022b)

Type locality: Holotype: BPBM C1599, collected on wet tables with flow through seawater at the Hawai‘i Institute of Marine Biology (HIMB) on Moku o Lo‘e (Coconut Island), Kāne’ohe Bay, O’ahu (21.4335°N, −157.78634°W), 1–3 m, collected on 2017-04-28, coll. Jan Vicente; Paratypes: BPBM C1604, UF 3973/BPBM C1601, BPBM C1602 collected on 2017-06-07; UF 3972/BPBM C1608, BPBM C1603, UF 3974/BPBM C1614 collected on 2017-09-27; BPBM C1616, BPBM C1619, BPBM C1620, BPBM C1618, BPBM C1615, BPBM C1621, BPBM C1617 collected on 2018-06-11; BPBM C1609, BPBM C1610 collected on 2017-02-13; BPBM C1611, BPBM C1605 collected on 2018-03-16; BPBM C1607, collected on 2017-08-01; BPBM C1613, BPBM C1612 collected on 2016-12-19; BPBM C1606; BPBM 1735 collected on 2017-11-21, from ARMS in mesocosms at HIMB in Moku o Lo‘e, Kāne‘ohe Bay, O‘ahu, (21.4335°N, −157.78634°W), 0.3 m. UF 3840/BPBM C1600, collected on the south side of Mokoli‘i Island (21.5085°N, −157.8295°W) living under coral rubble on fringing reef flat; 1–4 m deep; collected on 2017-06-01. BPBM C1651, BPBM C1652 collected on 2022-08-04; BPBM C1653, BPBM C1654, BPBM C1655, BPBM C1656 collected on 2022-09-26; BPBM C1657, BPBM C1658, BPBM C1659, BPBM C1660 collected on 2022-09-27 from ARMS after a 6-year deployment on the Northeast side of Moku o Lo‘e, Kāne‘ohe Bay, O‘ahu (21.4360°N, −157.7867°W); 0.1–1 m. The collector for paratypes is the same as the holotype.

Diagnosis: Stelletta hokunalohia sp. nov. lacks microscleres and has a wide range of intraspecific color variation ranging between white, tan, and different shades of purple which makes it unique among other Stelletta species.

Description (Fig. 3): Pedunculate (2 cm wide × 3 cm in height with a stem 0.5 cm thick) (Fig. 3A), globular (1–4 cm in diameter) (Figs. 3B–3D), but mostly irregular-shaped (Figs. 3E, 3F). The surface is bumpy, rugose, or rough with a firm consistency. Oscula (1.59 mm in diameter) are slightly raised and observed on most individuals. The color of the cortex of live specimens varies between white, tan, and different shades of purple or burgundy. Some individuals may show concurrent variation of these colors. The color of the choanosome is consistently cream. The tissue surrounding the oscula has a blotchy color pattern of brown, white, or beige. In ethanol, both the cortex and choanosome are white, beige, or light gray. In samples that had a dark purple exterior, a green pigment was released when fixed in ethanol.

Skeleton (Figs. 4A–4C): The cortex (277–469–730 μm in length) is made up of radially organized plagiotriaenes and anatriaene clusters that form bouquets (Figs. 4A, 4B). Distinct bouquets are seen divided by subcortical cavities (81–195–408 μm in diameter) (Figs. 4B, 4C). The choanosomal skeleton is composed of densely packed oxeas creating a separate layer from the cortex. Oxeas, anatriaenes (rare), and plagiotriaenes are oriented radially and diagonally to plagiotriaenes and anatriaenes in the cortex to support the choanosomal layer.

Spicules: (Figs. 4D–4J, Table 2): Plagiotriaenes (293–509–696 × 6–19–31 μm) have a smooth rhabdome with pointed tips (Fig. 4G); cladome (30–124–178 μm in diameter) is smooth with pointed tips (Fig. 4D). Anatriaenes (294–445–636 × 5–9–12 μm) have smooth, slender, rhabdomes with pointed tips (Fig. 4H); cladomes (28–38–53 μm in diameter) are smooth with pointed tips (Fig. 4E). Oxeas (421–636–861 × 7–13–18 μm) are smooth and fusiform (Fig. 4I). Some oxeas are slightly curved in the middle but were rare. A single, deformed megasclere (633 × 5 μm) was observed (Figs. 4F and 4J) which appeared as if two plagiotriaene cladomes (measuring 22 and 25 μm in diameter) were separated by a distance of 43 μm by a common rhabdome which tapers to sharply pointed opposite ends.

Habitat and ecology: Specimens were collected within natural and artificial sciophilous communities of Kāne‘ohe Bay including a piece of dead coral from a patch reef, ARMS units embedded in a shallow reef habitat, and ARMS in mesocosms.

Distribution: Kāne‘ohe Bay, O‘ahu, Hawai‘i between 1–4 m depth.

Etymology: “hokunalohia” is derived from two Hawaiian words: hōkū (star) and nalohia (lost or missing), translating to “lost star”. This name was chosen to reflect the lack of microscleres in this species.

Taxonomic remarks: Similar to Stelletta kela sp. nov., Stelletta hokunalohia sp. nov. resembles Stelletta the most by having abundant triaenes and oxeas. The absence of microscleres in Stelletta hokunalohia sp. nov. is a character only shared with Stelletta anasteria Esteves & Muricy, 2005 among 150 Stelletta spp. Although the absence of microscleres is a character shared between Stelletta hokunalohia sp. nov. and Stelletta anasteria, the latter has much smaller plagiotriaenes (210–491 × 2–9 μm), smaller plagiotriaene cladomes (23–95 μm), smaller anatriaene rhabdome widths and cladomes (2–5 and 11–25 μm), and smaller oxea (242–559 × 2–8 μm). Aside from spicule composition, other morphological characters also distinguish Stelletta hokunalohia sp. nov. For example, Stelletta anasteria is pale in color, has no visible oscules, and spherulous cells are present in the skeleton. Stelletta hokunalohia sp. nov. can be white, tan, or different shades of purple or burgundy, has very distinct, visible oscules, and spherulous cells are absent in the skeleton.

From the tetractinellids already described for Hawai‘i (Table S2), Stelletta debilis (sensu: de Laubenfels, 1951) is distinguished from Stelletta hokunalohia sp. nov. by the presence of oxyeuasters in Stelletta debilis.

Stelletta kuhapa sp. nov. (Figs. 5, 6, Table 3) urn:lsid:zoobank.org:act:776D40C2-33A9-41D5-AD89-108C75362C42.

Figure 5 In situ photos of Stelletta kuhapa sp. nov.

(A) Holotype (UF 3978/BPBM C1666) with arrow indicating Stelletta kuhapa sp. nov.; (B) paratype (UF 3975/BPBM C1661); (C) paratype (BPBM C1687); (D) paratype (UF 3977/BPBM C1664); (E) paratype (BPBM C1662); (F) paratype (BPBM C1684).

Figure 6 Skeleton and spicule composition of Stelletta kuhapa sp. nov.

(A) Cross section of holotype (UF 3978/BPBM C1666); (B) cross section of paratype (UF 3975/BPBM C1661); (C) zoomed in cross section of paratype (UF 3975/BPBM C1661); (D–K) light microscopy and SEM of spicules from holotype; (D) oxea; (E) forked oxea from light microscopy; (F) plagiotriaene; (G) anatriaene; (H) zoomed in anatriaene cladome; (I) zoomed in plagiotriaene cladome; (J, K) tylasters. Black arrows point to subcortical cavities (sc) and choanosomal spaces (cs) in (A) and (B).

Table 3 Spicule measurements for Stelletta kuhapa sp. nov.

Voucher	Anatriaenes	Plagiotriaenes	Oxeas	Microscleres	
	Rhabdome	Cladome	Rhabdome	Cladome		Tylasters	
	Length × Width	Diameter	Length × Width	Diameter	Length × Width	Diameter	
UF 3978/BPBM C1666 (Holotype)	443.5–518–581.7 × 8.7–15–19.8	39.5–52.5–62.7	410.3–573.2–678.3 × 16.5–26.9–35.8	119.4–180.6–243.2	452.8–778.1–884.2 × 16–26.6–33.5	8.2–9.7–11	
UF 3975/BPBM C1661	488–630.9–764.9 × 10.1–13.2–16.7	32.5–43.4–51.3	276.8–499.9–655.3 × 10.4–22–33.3 (n = 17 length)	59.3–154–246.9 (n = 6)	605.9–834.6–988.0 × 11.2–22.2–30	7.3–8.9–10.6	
BPBM C1662	398–539.5–640.3 × 6.8–11.3–13.7	34.7–39.1–48.6	376–583.4–740.2 × 15.5–21.4–27	96–249.4–317.9 (n = 6)	524.1–758.5–948.3 × 8.8–14.9–21.6	5.6–6.7–8	
UF 3977/BPBM C1664	394–483.7–554.4 × 11.8–15.1–17.3	34.6–50.7–57.5	483.1–562.9–675.1 × 14.2–27.5–35.4	134–192.3–216.5	598.5–749.3–864.8 × 11.7–23.3–29.6	–	
BPBM C1687	416.8–484.8–645.7 × 11–14.4–19.7	37.3–45.5–58.8	275.1–496.9–639.8 × 14.2–22.7–33.1	92.8–160.9–213.1	578.5–759.5–933.7 × 12.7–20–25.3	–	
BPBM C1684	385.8–489.5–585 × 7.8–13–16.6	27.3–46.6–57.8	290.6–477.5–631 × 9.2–18.2–27.1	93.9–165.3–215.8	577.8–771.1–852.9 × 15.5–21.7–27.4	–	
Note:

Measurements shown as minimum—average—maximum. n = 30 for megascleres and n = 15 for microscleres unless otherwise noted. Microscleres were not measured for all vouchers because they were rare and only visible in cross sections. Cross sections were made for three vouchers. All measurements are presented in μm.

Ancorinidae sp. 5 in Vicente et al. (2022b) and Ancorinidae sp. 7 in Vicente et al. (2022a, 2022b)

Type locality: Holotype: UF 3978/BPBM C1666, collected from ARMS in mesocosms at the Hawai‘i Institute of Marine Biology (HIMB) in Moku o Lo‘e (Coconut Island), Kāne‘ohe Bay, O‘ahu, (21.4335°N, −157.7864°W), 0.3 m, collected on 2017-09-27, coll. Jan Vicente. Paratypes: BPBM C1663, UF 3977/BPBM C1664, UF 3976/BPBM C1665 collected on 2017-09-27; BPBM C1662 collected on 2017-02-13; BPBM C1667, UF 3975/BPBM C1661 collected on 2017-08-11; BPBM C1670, BPBM C1671, BPBM C1684, BPBM C1685 collected on 2018-03-16; BPBM C1687, BPBM C1688 collected on 2018-06-11. Location, collector, and depth for all paratypes are the same as the holotype.

Diagnosis: Stelletta kuhapa sp. nov. is distinguished from other Stelletta species by its spicule composition (plagiotriaenes, anatriaenes, oxeas, and tylasters) and spicule size, as well as its intraspecific external color variation which ranges between light tan, light brown, and dark burgundy.

Description (Fig. 5): Thickly encrusting (1–6 cm), irregularly shaped (0.5 cm) sponge. Surface is rugose and bumpy with a tough consistency. One to multiple osculum (0.5 mm in diameter) visible on each specimen elevated by a translucent membrane. The color of live specimens varies between tan, white, beige, grey, light brown, and dark burgundy. The color of the sponge interior is cream. In ethanol, the cortex and choanosome are white or beige.

Skeleton (Figs. 6A–6C): The cortex (446–569–730 μm in length) is made up of radially organized plagiotriaene and anatriaene clusters (bouquets) separated by subcortical cavities (135–298–679 μm in diameter) (Figs. 6A, 6B). The choanosomal skeleton is primarily composed of densely packed plagiotriaenes and anatriaenes creating a separate layer from the cortex. Oxeas, anatriaenes, and plagiotriaenes create a scattered, confused choanosome and are oriented radially and diagonally to the plagiotriaenes and anatriaenes of the cortex. Choanosomal spaces (309–380 μm in diameter) can be seen in some skeleton sections located just below the choanosomal layer (Fig. 6A). Tylasters are abundant and are concentrated in the cortex.

Spicules (Figs. 6D–6K, Table 3): Plagiotriaenes (275–532–740 × 9–23–36 μm) have a smooth, thick rhabdome with pointed tips, (Fig. 6F). Occasionally, plagiotriaenes showed rounded tips but were uncommon. Cladome (59–184–318 μm in diameter) is smooth with pointed tips (Fig. 6I). Anatriaenes (385–524–765 × 6–14–20 μm) have smooth, thick rhabdomes with pointed tips (Fig. 6G). Some rhabdomes had thorns and some had rounded tips, but were uncommon. Cladomes (27–46–63 μm in diameter) are smooth with pointed tips and short clads (Fig. 6H). Oxeas (452–775–988 × 8–22–34 μm) are smooth, thick, and blunt (Fig. 6D). Some oxeas are slightly curved in the middle, and some have forked tips, but these were rare (Fig. 6E). Tylasters, abundant, approximately 6–11 rays, 5–8–11 μm in diameter (Figs. 6J, 6K). Styles were observed occasionally but were uncommon.

Habitat and ecology: Specimens were collected from ARMS inside mesocosms supplied with unfiltered flow through seawater at the Hawai‘i Institute of Marine Biology (HIMB) in Moku o Lo‘e (Coconut Island). During the same period specimens were not observed with ARMS on a reef surrounded by a climax sponge community throughout a 2-year monitoring period (Sup. Fig. S3 in Vicente et al., 2022a). This suggests that these species are pioneering during community development because they only appeared in habitats completely removed from a fully developed coral reef ecosystem. These species seem to be strictly sciophilous and have yet to be found on natural calcified surfaces.

Distribution: Moku o Lo‘e, Kāne‘ohe Bay, O‘ahu, Hawai‘i, 0.3 m depth.

Etymology: The Hawaiian word “kūhapa” means to vary in size or appearance. This name was chosen to reflect the wide intraspecific color variation of this species.

Taxonomic remarks: The presence of plagiotriaenes, anatriaenes, oxeas, and euasters places this sponge within the family Ancorinidae. Similarly to Stelletta kela sp. nov. and Stelletta hokunalohia sp. nov., Stelletta kuhapa sp. nov. has abundant triaenes, oxeas, and euasters which resembles other Stelletta congeners. Of the 18 species that had matching spicule compositions (Table S3), only Stelletta beae Hajdu & Carvalho, 2003 and Stelletta paucistellata (Lévi, 1952) have spicule sizes within the limits of Stelletta kuhapa sp. nov. Although similar in size, Stelletta beae has orthotriaenes (87–737 × 2–41 μm; cladome, 19–310 μm in diameter) and smaller anatriaene cladomes (7–30 μm). Stelletta paucistellata has smaller plagiotriaene cladomes (65–135 μm), smaller anatriaene cladomes (25 μm), and is found near Western Africa.

From the tetractinellids already described in Hawai‘i (Table S2), Stelletta debilis (sensu: de Laubenfels, 1951) is distinguished by smaller plagiotriaene cladomes (32 μm) and the presence of oxyeuasters which are absent in Stelletta kuhapa sp. nov.

Stelletta hokuwanawana sp. nov. (Figs. 7, 8, Table 4) urn:lsid:zoobank.org:act:1EF50A29-2EE8-491E-B783-5A3A5D7666A7.

Figure 7 In situ photos of Stelletta hokuwanawana sp. nov.

(A) Holotype (BPBM C1634) with white arrow indicating oscula; (B) paratype (BPBM C1691); (C) paratype (BPBM C1630); (D) paratype (BPBM C1633).

Figure 8 Skeleton and spicule composition of Stelletta hokuwanawana sp. nov.

(A) Cross section of holotype (BPBM C1634); (B) cross section of paratype (BPBM C1633) with a black arrow point to a choanosomal space (cs); (C) zoomed in cross section of holotype; (D, E) zoomed in cross section showing spherulous cells from paratype (BPBM C1633); (F–L) SEM of spicules from holotype; (F) forked oxea from light microscopy; (G, I) oxea; (H) style; (J–L) acanthospherasters. Spherulous cells are indicated by white arrows on (B), (D, E).

Table 4 Spicule measurements for Stelletta hokuwanawana sp. nov.

Voucher	Oxeas	Acanthospherasters	
	Length × Width	Diameter	
BPBM C1634 (Holotype)	777.9–916.8–1069.2 × 26.6–40.5–51.1	6–7.8–8.9	
BPBM C1630	535.5–782.2–946.7 × 13.5–25.3–40.8	6–7–8.4	
BPBM C1691	612–771.1–948.1 × 14.2–29–45.4	6.1–7.5–9.2	
BPBM C1633	700–909.3–1,089.7 × 17.8–34.5–50.5	6.2–7.8–9	
Note:

Measurements shown as minimum—average—maximum. n = 30 for megascleres and n = 10 for microscleres unless otherwise noted. All measurements are presented in μm.

Type locality: Holotype: BPBM C1634 collected from ARMS embedded in the reef on the Northeast side of Moku o Lo‘e, Kāne‘ohe Bay, O‘ahu (21.4360°N, −157.7867°W), 0.1–1 m; coll. Jan Vicente, collected on 2022-05-10. Paratypes: BPBM C1633, BPBM C1630 collected on 2023-05-30; BPBM C1691 collected on 2022-09-26. Location, depth, and collector for all paratypes are the same as the holotype.

Diagnosis: Stelletta hokuwanawana sp. nov. is the only Stelletta species to exhibit an encrusting form, with a range in color between white, beige, dark grey and light grey, and an absence of triaenes.

Description (Fig. 7): Thin to thick, irregularly shaped encrustation (2–12 × 0.6–2 cm), with a hispid surface from protruding megascleres. Surface is even, consistency is tough, making it difficult to squeeze. Small oscula (1 mm in diameter) can be seen on the top of some individuals encrusting with slender morphology (Fig. 7A). The color of the cortex varies between dark grey, white, and beige. The choanosome is tan in color. In ethanol, the cortex is dark grey and the choanosome is white.

Skeleton (Figs. 8A–8E): The cortex (599–698–783 μm in length) is formed by a dense, confused layer of oxeas oriented diagonally and perpendicular to the cortex surface. The choanosomal skeleton is composed of densely packed oxeas oriented perpendicular and radially to the cortex layer. Some cross-sections show choanosomal spaces (224–519–1,331 μm in diameter) (Fig. 8B) as well as spherulous cells (32–42–50 μm in diameter) which are dark and round in appearance and are abundant throughout the cortex and choanosome (Figs. 8B, 8D, 8E). Acanthospherasters are scattered throughout the cortex and choanosome.

Spicules (Figs. 8F–8L, Table 4): Oxeas (536–845–1,090 × 14–32–51 μm) are smooth, fusiform with acerate tips (Figs. 8F, 8G, 8I). Two styles (817 × 42 μm) were found but were rare (Fig. 8H). Deformed oxeas (e.g., forked tips (Fig. 8F), bumpy rhabdomes, horns) were also rare. A single monaene was also observed. Acanthospherasters (6–8–9 μm in diameter), consisted of approximately 15–20 rays, with a thick centrum exceeding the length of the arms. All arms are short, blunt, and spiked with a somewhat smooth centrum if visible (Figs. 8J–8L). Occasionally rays were longer than the centrum with about 5–10 rays, but this was uncommon (Fig. 8L).

Habitat and ecology: Specimens were collected from ARMS embedded in a shallow reef after a 6-year recruitment period. These species seem to be strictly sciophilous and have yet to be found on natural substrates.

Distribution: Moku o Lo‘e, Kāne‘ohe Bay, O‘ahu, Hawai‘i between 0.1–1 m depth.

Etymology: “hokuwanawana” is derived from two Hawaiian words: hōkū (star) and wanawana (spiny or thorny), translating to “thorny star”. This name was chosen to reflect the spiky microscleres in this species.

Taxonomic remarks: The presence of oxeas and euasters places the new species within the family Ancorinidae. Although the presence of triaenes is common among most ancorinids, several species belonging to different genera within Ancorinidae may lack triaenes. For example, Asteropus and Jaspis species are specifically defined as lacking triaenes. Stelletta hokuwanawana sp. nov. lacks triaenes but also lacks sanidasters which is a defining characteristic of Asteropus. Furthermore, Stelletta hokuwanawana sp. nov. has euasters with a very clear centrum which are absent in Jaspis spp. (Uriz, 2002).

Although Jaspis is characterized by euasters without a centrum, one species, J. grisea Lévi, 1959, shares some similarities in spicule composition and external coloration with Stelletta hokuwanawana sp. nov. However, J. grisea has slightly smaller oxeas (450–700 μm), is massive in form, and is found off the west coast of Africa making it an unlikely conspecific to Stelletta hokuwanawana sp. nov. Given these differences, Stelletta resembles this new species the most due to the abundance of euasters with a thick centrum that are found throughout the cortex and choanosome.

Within Stelletta, eight species match Stelletta hokuwanawana sp. nov. in spicule composition by only having oxeas and euasters (Table S3). Of these eight species, two fall within the size limits of Stelletta hokuwanawana sp. nov., Stelletta pudica (Wiedenmayer, 1977) and Stelletta tuberculata (Carter, 1886). Although similar in size, Stelletta pudica is subspherical in shape, has a yellow, red, or purplish-brown color, thinner oxeas (8–15 μm), and tylasters (12–13 μm) which are absent in Stelletta hokuwanawana sp. nov. Stelletta tuberculata (sensu: de Laubenfels, 1954) is globular in shape, yellow-brown or pink in color, tuberculate, lumpy, has smaller euasters (4–5 μm), and slightly thinner oxeas (13–33 μm).

Of the twelve tetractinellid species already identified in Hawai‘i (Table S2), Jaspsis digonoxea, Asteropus kaena, and Jaspsis pleopora all contain similar morphological characters but still some key morphological differences stand out. Jaspis digonoxea has rare, smaller oxeas (400–520 × 7–12 μm) and the presence of microxeas and oxyeuasters which are absent in Stelletta hokuwanawana sp. nov. Asteropus kaena has larger oxeas (1,000–2,400 × 14–42 μm) and the presence of oxyeuasters and streptasters (12–20 μm in diameter) which are absent in Stelletta hokuwanawana sp. nov. Jaspis pleopora has smooth spherasters (7–20 μm in diameter) and the presence of oxyeuasters (10 μm in diameter) which are absent in Stelletta hokuwanawana sp. nov.

Stelletta apapaola sp. nov. (Figs. 9, 10, Table 5) urn:lsid:zoobank.org:act:D203AB34-AA4C-4DFA-AB6D-1610A4FE4284.

Figure 9 In situ photos of Stelletta apapaola sp. nov.

(A) holotype (BPBM C1632) with a white arrow indicating oscula; (B) paratype (BPBM C1631); (C) paratype (BPBM C1694); (D) paratype (BPBM C1692); (E) paratype (BPBM C1693). White arrows indicate subsurface channels (B, D).

Figure 10 Skeleton and spicule composition of Stelletta apapaola sp. nov.

(A) Cross section of paratype (BPBM C1631); (B) cross section of holotype (BPBM C1632); (C) zoomed in cross section of holotype with a white arrow indicating a thin layer of spherasters creating an ectosome; (D–J) SEM of spicules from holotype; (D) style; (E, G, H) oxea; (F) monaene; (I) oxyspheraster; (J) acanthospheraster.

Table 5 Spicule measurements for Stelletta apapaola sp. nov.

Voucher	Oxeas	Styles	Microscleres	
	Length × Width	Length × Width	Diameter	
BPBM C1632 (Holotype)	522.5–660.5–793.2 × 10.2–18.8–23.5	565.9–616.3–709.9 × 17–21.8–25.3 (n = 9)	9.3–11.1–15.8	
BPBM C1631	424.1–666.5–851.5 × 10.8–27.9–41.9	787.8 × 35.7 (n = 1)	11.7–13.4–15.8	
BPBM C1693	473.1–707.3–899.5 × 16.9–23.8–31	518.7 × 17.3 (n = 1)	10.8–12.8–15	
BPBM C1694	472.6–723.3–822.4 × 13–24.6–30.4	Absent	9.7–12.5–15	
Note:

Measurements shown as minimum—average—maximum. n = 30 for megascleres and n = 10 for microscleres unless otherwise noted. All measurements are presented in μm. Both oxyspheraster and acanthospheraster measurements were grouped due to similarities in size and the inability to tell them apart unless observed under SEM.

Type locality: Holotype: BPBM C1632, collected from ARMS embedded in the reef on the Northeast side of Moku o Lo‘e, Kāne‘ohe Bay, O‘ahu (21.4360°N, −157.7867°W), 0.1–1 m, coll. Jan Vicente, collected on 2023-04-20. Paratypes: BPBM C1631, collected on 2023-05-30; BPBM C1693, BPBM C1694, collected on 2022-08-04; BPBM C1692, collected on 2022-09-27. Location, depth, and collector for all paratypes are the same as the holotype.

Diagnosis: Stelletta apapaola sp. nov. is thinly encrusting, grey or white in color, and lacks triaenes which comprehensively is unique, only to this species.

Description (Fig. 9): Thinly (3–5 mm thick) encrusting spreading (2–4 × 0.2–1 cm) irregularly with an even surface. The consistency of the sponge is firm. Oscula (2.1 mm in diameter) are rare, observed on one individual, and is slightly elevated (Fig. 9A). Subsurface channels (348 μm wide) span the length of the sponge surface (Figs. 9B, 9D). The color of live specimens vary between white and light grey. The choanosome is tan in color. In ethanol, the specimen is grey or white with a white interior. In some specimens, there is no visible distinction between cortex and choanosome.

Skeleton (Figs. 10A–10C): The ectosome is not clearly defined from the choanosome. The choanosomal skeleton is composed of oxeas oriented perpendicularly and diagonally to the sponge surface. One choanosomal space (213 μm in diameter) was observed. Acanthospherasters and oxyspherasters are abundantly scattered throughout the sponge body (Fig. 10C).

Spicules (Figs. 10D–10J, Table 5): Oxeas (424–689–900 × 10–24–42 μm) are smooth, fusiform (Figs. 10E, 10G, 10H). Styles (519–641–788 × 17–25–36 μm) were rare (Fig. 10D). Occasional deformed oxeas (e.g., forked tips (Fig. 10G), bumpy rhabdomes, horns) were rare. A single monaene (Fig. 10F) was observed. Oxyspherasters consisted of approximately 15–20 rays and a thick, smooth centrum (10–13–16 μm in diameter) (Fig. 10I). Each ray was smooth except for small, infrequent spikes located at the tips. Acanthospherasters (9–12–13 μm in diameter) consisted of approximately 15–20 rays with a thick centrum exceeding the length of the rays (Fig. 10J). All arms are spiked and short with a smooth centrum.

Habitat and ecology: Specimens were collected from ARMS embedded in a shallow reef after a 6-year recruitment period. These species seem to be strictly sciophilous and have yet to be found on natural substrates.

Distribution: Moku o Lo‘e, Kāne‘ohe Bay, O‘ahu, Hawai‘i between 0.1–1 m depth.

Etymology: “apapaola” is derived from two Hawaiian words: ‘āpapa (meaning coral reef flat) and ola (meaning life), translating to “reef life”. This name was chosen to emphasize sponge life within the reef.

Taxonomic remarks: Similar to Stelletta hokuwanawana sp. nov., there is only one Jaspis species, J. grisea, that shares similarities with Stelletta apapaola sp. nov. However, J. grisea has slightly smaller oxeas (450–700 μm), smaller oxyasters (8 μm in diam.), is massive in form, and is found off the west coast of Africa making it an unlikely conspecific to Stelletta apapaola sp. nov. This species is more closely related to other Stelletta congeners, as indicated by the size and shape of euasters. Stelletta hokuwanawana sp. nov. and Stelletta apapaola sp. nov. share some similarities although they are described as different species due to Stelletta hokuwanawana sp. nov. lacking oxyspherasters and having much smaller acanthospherasters (6–9 μm in diameter).

Of the eight Stelletta species that share similar spicule compositions (i.e., oxea and euasters) (Table S3), only Stelletta tuberculata and Stelletta jonesi Thomas, 1973 stand out as close relatives to Stelletta apapaola sp. nov. Although similar in spicule composition, Stelletta tuberculata (sensu: de Laubenfels, 1954) has slightly larger oxeas with thinner widths (900–1,155 × 13–18 μm), much smaller euasters (4–5 μm in diameter), is yellow-brown or pink in color, lumpy, and tuberculate. Stelletta jonesi has much bigger oxyasters (50–96 μm), lacks acanthospherasters, and is found near Seychelles Bank.

Of the twelve Hawaiian tetractinellids already described (Table S2), only Jaspsis digonoxea, Asteropus kaena, and J. pleopora share similar spicule compositions. However, J. digonoxea has rare, smaller oxeas (400–520 × 7–12 μm) and the presence of microxeas (105 × 3 μm) which are absent in Stelletta apapaola sp. nov. Asteropus kaena has larger oxeas (1,000–2,400 × 14–42 μm) and the presence of streptasters (12 μm) which are absent in Stelletta apapaola sp. nov. Jaspis pleopora has much thinner oxea (6–8 μm), a wider range of sizes in spherasters and oxyeusaters (7–20 μm), as well as oxyeuasters that have a fewer number of rays (6–8).

Genus Stryphnus Sollas, 1886

Stryphnus huna sp. nov. (Fig. 11, Table 6) urn:lsid:zoobank.org:act:91AB648C-8712-439D-94AF-26E1689F9657.

Figure 11 In situ photos, skeleton, and spicule composition of Stryphnus huna sp. nov. (BPBM C1690).

(A) In situ; (B) cross section; (C) zoomed in of cross section; (D–L) SEM of spicules; (D–E) oxyasters; (F, I) styles; (G, H, J) oxeas; (K–L) sanidasters.

Table 6 Spicule measurements for Stryphnus huna sp. nov.

Voucher	Oxea	Microscleres	
	I: length: <1,000 um, width <30 um	II: length: >1,000 um, width: ≥30 um	Sanidasters	Oxyasters	
	Length × Width	Length × Width	Length	Diameter	
BPBM C1690 (Holotype)	406.0–748.9–967.8 × 7.5–15.5–23.6	1,148.8–1,473.5–1,861.3 × 38.3–49.9–59.9	8.8–12.6–16.6	13.9–36.2–51 (n = 6)	
Note:

Measurements shown as minimum—average—maximum. n = 30 for megascleres and n = 15 for microscleres unless otherwise noted. All measurements are presented in μm.

Ancorinidae sp. 8 in Vicente et al. (2022b)

Type locality: Holotype: BPBM C1690, collected from ARMS on reef at the Hawai‘i Institute of Marine Biology (HIMB) in Moku o Lo‘e (Coconut Island), Kāne‘ohe Bay, O‘ahu, (21.4334°N, −157.7868°W), 3 m, collected on 2018-06-11, coll. Jan Vicente.

Diagnosis: Stryphnus huna sp. nov. is characterized by its encrusting growth form, spicule composition (oxeas in two sizes, sanidasters, and oxyasters), and spicule dimensions.

Description (Fig. 11A): Thinly (2 mm thickness) encrusting sponge that spread laterally (4 × 2 cm). No oscula visible. Surface is somewhat rubbery to hispid. Consistency is tough and difficult to tear. The color of the live and fixed specimen in ethanol is a light greyish-brown.

Skeleton (Figs. 11B, 11C): The cortex is not specialized or clearly defined from the choanosome. The choanosome is formed by densely packed oxeas (of both sizes) oriented radially and diagonally. Sanidasters are abundant in the sponge body. Oxyasters are rare and scattered throughout the choanosome.

Spicules (Figs. 11F–11L, Table 6): Oxeas were observed in two size categories (I: 406–749–968 × 8–16–24 μm and II: 1,149–1,474–1,861 × 38–50–60 μm), are massive, smooth, fusiform, somewhat flexuous, or slightly curved but not at the center (Figs. 11G, 11H, 11J). The smaller oxeas are thin and also slightly curved. Styles measure 1,172 × 51 μm and are rare (Figs. 11F, 11I). Occasionally, forked oxeas and styles were seen, but are also rare (Fig. 11F). Sanidasters, spined, with approximately 10–20 arms coming out of the main shaft measuring 9–17 μm in length (Figs. 11K, 11L). Oxyasters (Fig. 11D and 11E) are extremely rare, with 3–10 rays, measuring 14–51 μm in diameter.

Habitat and ecology: Specimens were collected from ARMS embedded in a shallow reef after a 2 year recruitment period. These species seem to be strictly sciophilous and have yet to be found on natural calcified surfaces.

Distribution: Moku o Lo‘e, Kāne‘ohe Bay, O‘ahu, Hawai‘i, 3 m depth.

Etymology: “hūnā” is a Hawaiian word which means to hide. This word was chosen for the cryptic nature and rarity of this species.

Taxonomic remarks: The presence of oxeas, sanidasters, and oxyasters places this sponge within the family Ancorinidae. This new species falls within two genera of Ancorinidae, Stryphnus and Asteropus. Within Stryphnus, only Stryphnus radiocrusta (Kennedy, 2000) shares a similar spicule composition and falls within the size range of the new species. Stryphnus radiocrusta has slightly smaller oxyasters (8–25 μm), is massive, subspherical, and has a clearly defined cortex.

Within Asteropus, eight species have similar spicule composition (Table S3). Of those eight, only A. simplex (Carter, 1879) and A. syringifer van Soest & Stentoft, 1988 share a similar geographic range, spicule composition, and size with the new species. The type specimen of A. simplex has smaller oxea (1,270 × 25 μm), similar oxyasters (20–50 μm), and similar sanidasters (15 μm). Paratypes of A. simplex ranges in oxea sizes between, 130–2,915 × 3–81 μm, oxyasters, 3–12 rays, 14–58 μm, and sanidasters 10–25 μm (Carter, 1879; Bergquist, 1968, 1969; van Soest, 1981b; Desqueyroux-Faúndez, 1990). However, the type and paratypes are either massive encrustations or convex-shaped cushions with a clearly specialized ectosome. The new species is only 2 mm thick and the ectosome is not specialized. Asteropus syringifer is also a massive, round-shaped sponge with a specialized ectosome. Spicule composition is somewhat similar to the new species by having slightly thinner oxea (15–45 μm), slightly bigger sanidasters (12–20 μm), and slightly larger oxyasters (40–60 μm).

As published in Cárdenas et al. (2011), Asteropus is proposed as a junior synonym of Stryphnus, therefore this new species is placed within Stryphnus following molecular data (see phylogenetic tree and analysis below for further explanation).

Family Geodiidae Gray, 1867

Subfamily Geodiinae Gray, 1867

Genus Geodia de Lamarck, 1815

Geodia cf. papyracea Hechtel, 1965

(Figs. 12, 13, Table 7).

Figure 12 In situ photos of Geodia cf. papyracea.

(A) (BPBM C1626) With an arrow showing single osculum; (B) (BPBM C1737) with arrows indicating cribriporal surface; (C) ectosome and cortex distinction indicated by arrow (BPBM C1737); (D) (BPBM C1736) with arrows showing oscular plate; (E) oscular plate with uniporal oscules (BPBM C1736).

Figure 13 Skeleton and spicule composition of Geodia cf. papyracea.

(A) Cross section of BPBM C1626 with reproductive elements indicated by black arrows; (B) zoomed in cross section (BPBM C1626); (C) zoomed in cross section with reproductive elements showing spicules inside; (D–L) SEM of spicules from BPBM C1626; (D) sterrasters with warty rosettes and hilum; (E) zoomed in sterraster surface of warty rosettes; (F) acanthoxyaster; (G) acanthostrongylaster; (H) anatriaene; (I) oxea II; (J) plagiotriaene; (K) monaene; (L) oxea I.

Table 7 Spicule measurements for Geodia cf. papyracea.

Voucher	Anatriaenes	Plagiotriaenes	Oxea	Monaene	Microscleres	
	Rhabdome	Cladome	Rhabdome	Cladome	I: 100–400 µm	II: 600–1,100 µm	Rhabdome	Sterrasters	Acanthoxyaster	Acanthostrongylaster	
	Length × Width	Diameter	Length × Width	Diameter	Length × Width	Length × Width	Length × Width	Diameter	Diameter	Diameter	
BPBM C1626	rare, 835.8–889.5–948.7 × 6.6–7.3–8.2 (n = 3)	36.8–40.9–46.3 (n = 3)	622.8–814–1,001.8 × 11–16.8–24.6	62.8–139.1–206.3	151.9–306.5–396.3 × 1.5–3.3–6.2	673.8–887.3–1,257.8 × 13.6–19.3–23.3	760.6–804.9–849.2 × 12.8–14.3–15.7 (n = 2)	36.1–47.6–57.8	15.7–23.1–29.6 (n = 20)	6.8–7.5–8.4 (n = 10)	
BPBM C1736	None	None	718.2 × 17.3 (n = 1)	205.5 (n = 1)	None	730.3–953.7–1,111.5 × 15–20.5–26.3	822 × 22.5 (n = 1)	19.8–38.2–52 (n = 20)	33–37.5–43.3 (n = 7)	4.3–5–6.1 (n = 3)	
BPBM C1737	None	None	588.5–835.5–1,017.9 × 7.9–18.3–26.6 (n = 15)	49.1–138.1–220.5 (n = 10)	178.1–205–231.9 × 3.4–3.7–4 (n = 2)	799.7–1,012.8–1,154.2 × 14.5–22.4–27.7	None	30.2–43–52.5 (n = 20)	24.6–33.3–40.1 (n = 19)	3.2–4.5–5.7 (n = 17)	
Geodia papyracea, YPM 5045, Jamaica (Hechtel, 1965)	246–609 × 2–3	7–36	406–1,058 × 5–24	33–123	–	616–1,183 × 9–20	–	52–75	6–12 thin, roughened rays, with or without a small centrum, 24–48	8–10 blunt rays, 5–7	
Geodia papyracea, UMPCW921, Panama (Cárdenas et al., 2009)	Rare, 354–633.2–885 × 2.5–3.2–4	Clad length: 5–13.8–27	641–962.7–1,080 × 13–24.6–34	57–97.1–132	95–124.3–244 × 1–2–3	651–1039–1248 × 7–24.3–29	–	4–7 Branched rosettes, 63–70.3–75	I: 22–27.5–36; II: 14–23–29	14–20 Actines,
3–4.9–8.8	
Note:

Measurements shown as minimum—average—maximum. n = 30 unless otherwise noted. All measurements are presented in μm. Specimens from this study are in bold.

Synonyms and reference: Geodia (Cydonium) papyracea, Hechtel (1965): text-fig. 13, pl. VIII, figs.1–2; Geodia sp., Burton (1940): text pg 97–98; Tetractinellida sp. 1, Vicente et al. (2022b): Table S4 and S7; Cárdenas et al. (2009): fig. 16; Silva, Mothes & Lyrio-Oliveira (2004): fig. 2–28.

Type locality: YPM 5045, mangrove boat channel, Port Royal, Jamaica.

Comparative material from previous species descriptions: G. papyracea, YPM 5045, holotype, mangrove boat channel, Port Royal, Jamaica; YPM 5311, paratype, mangrove boat channel, Port Royal, Jamaica; UMPCW921, mangrove root, Solarte Island, Panama.

Material examined: BPBM C1626 collected on 2018-11-16, BPBM C1736 and BPBM C1737 collected on 2023-10-26. All specimens were collected on wet table outflow pipes at the Hawai‘i Institute of Marine Biology (HIMB) on Moku o Lo‘e (Coconut Island), Kāne‘ohe Bay, O’ahu (21.4334, −157.7873), <0.3 m, coll. Jan Vicente.

Description (Fig. 12): Globular, sub-globular (3–5 cm diameter) to irregular-shaped lobes (6–8 cm diameter). Consistency is firm and somewhat tough to tear. Surface of cortex is hispid and can be easily separated from the internal tissue. Oscules could be visible as a single osculum (2 mm in diameter) (Fig. 12A) or as a depressed oscular plate (8 mm in diameter) with multiple uniporal oscules (600 μm in diameter) (Figs. 12D, 12E). The color of the cortex from live specimens varies between white and purple but the choanosome is consistently tan. In ethanol, the choanosome and cortex are both tan or white. Cribriporal pores are distributed across the surface (Fig. 12B). In specimen BPBM C1626, the choanosome was mostly composed of reproductive elements (155–194–258 μm in diameter) which easily detach from the choanosome when preserved in ethanol, appearing as sediment in the sampling container (Fig. 13A).

Skeleton (Figs. 13A–13C): The cortex (266–534–761 μm thick) is composed of an ectocortex (0–66–262 μm thick) made up of acanthostrongylasters and an endocortex (167–327–629 μm thick) composed of sterrasters. Bouquets of oxeas and plagiotriaenes are positioned within the cortex so the head of the spicules align with the surface of the sponge. Oxeas, plagiotriaenes, and anatriaenes (rare) are positioned radially in the choanosome. Acanthoxyasters are scattered beneath the endocortex layer of sterrasters throughout the choanosome.

Spicules (Figs. 13D–13L, Table 7): Plagiotriaenes (622–789–1,018 × 7–18–27 μm) have a smooth rhabdome and cladome (49–161–221 μm in diameter) with pointed tips (Fig. 13J). Anatriaenes (836–890–949 × 7–7–8 μm) are rare, have a smooth rhabdome and cladome (37–41–46 μm in diameter) with pointed tips (Fig. 13H). Oxeas I (152–256–396 × 1–4–6 μm) are hair-like and can be curved, slightly curved, or straight (Fig. 13L). Oxeas II (674–951–1,258 × 14–21–28 μm) are smooth with pointed tips and slightly curved (Fig. 13I). Monaenes (761–814–849 × 13–18–23 μm) are rare, curved, and have pointed tips (Fig. 13K). Sterrasters (20–43–58 μm) with 5–7 branched warty rosettes (Figs. 13D, 13E). Acanthoxyasters (16–31–43 μm in diameter) with ~15–20 slender, spiked rays and a distinct centrum (Fig. 13F). Acanthostrongylasters (3–6–8 μm in diameter) have ~20 spiked, blunt rays and a thick centrum that exceeds the length of the rays (Fig. 13G).

Habitat and ecology: Specimens were found fouling the surface of outflow pipes of Moku o Lo‘e (Coconut Island), 0.3 m depth. Individuals have yet to be found on natural substrates.

Distribution: Jamaica (Hechtel, 1965); Colombia (Wintermann-Kilian & Kilian, 1984); Cuba (Alcolado, 2002); Belize (Rützler, 1988; Rützler et al., 2000); Panama (Díaz, 2005; Cárdenas et al., 2009); Curaçao (van Soest, 1981a); Brazil (Burton, 1940; Silva, Mothes & Lyrio-Oliveira, 2004; Cedro et al., 2007; Muricy et al., 2011; Bettcher et al., 2023); Hawai‘i (this study).

Taxonomic remarks: This is potentially the first record of Geodia cf. papyracea in Hawai‘i. The characteristics of these individuals match previously reported G. papyracea in the Caribbean. The Hawai‘i specimen and the holotype (YPM 5045, Hechtel (1965)) share similar color morphologies, cribriporal pores, and skeletal organization. However, there are some differences in spicule composition. The holotype has slightly bigger plagiotriaene rhabdomes (406–1,058 µm) but smaller cladomes (33–123 µm in diameter). The holotype also has smaller anatriaenes (246–609 × 2–3 µm; cladome: 7–36 µm in diameter). In the holotype, oxeas I are absent, and sterrasters (52–75 µm in diameter) are significantly bigger. Aside from these differences, the Hawaiian specimen and the holotype share morphological similarities (e.g., shape and size) with oxeas II, acanthoxyasters, and acanthostrongylasters.

Additionally, the Hawai‘i specimen shares similarities to the description of another Caribbean specimen from Panama (UMPCW921, Cárdenas et al. (2009)). Externally, the Panama and Hawai‘i specimens share few differences. The Hawai‘i specimen is white and purple while the Panama specimen is also white but has green tinges. Oscules were difficult to find in the Hawaiian specimen, likely due to the small size of the individuals in relation to the Panama specimen. However, some oscula were found as either a single opening or as an oscular plate with multiple pores on a depressed surface which match the description of the Panama specimen. The surface morphology of both specimens share the abundance of cribriporal pores. The skeleton of both specimens share similar skeletal organization and cortex thickness. However, the Panama specimen has a thinner ectocortex (0–72 µm) and a smaller size range of the endocortex (480–840 µm). Spicule composition and measurements share similarities but also notable differences. For example, the Panama specimen has smaller anatriaene rhabdomes and cladomes (354–633–885 × 3–3–4 μm, cladome 5–14–27 μm), slightly smaller plagiotriaene cladomes (57–97–132 μm), and slightly smaller oxeas I (95–124–244 μm). Additionally, the Panama specimen has bigger sterrasters than the Hawai‘i specimen (65–72–77 μm). Aside from these differences, characteristics and measurements of oxeas II, acanthoxyasters, and acanthostrongylasters match closely with the Panama specimen.

Phylogenetic analysis

The maximum-likelihood topology of the 28S rRNA gene places the new Stelletta and Stryphnus spp. within Clade A of Astrophorina (Cárdenas et al., 2011, Fig. 2). Species in Clade A consist of several families within Tetractinellida and in our analysis we include species belonging to Ancorinidae and Geodiidae (Fig. 14). This ‘Ancorinidae’ clade is nested within the Geodia clade and is hypothesized that the Geodia sp. have secondarily lost their sterrasters (Cárdenas et al., 2011, Fig. 2; Cárdenas, 2020, Fig. 9). Stelletta kela sp. nov., Stelletta hokunalohia sp. nov., and Stelletta kuhapa sp. nov. have triaenes (i.e., orthotriaenes, plagiotriaenes, anatriaenes), are all closely related genetically (95–97% identical), and are strongly supported by a monophyletic clade (bootstrap of 100) in both 28S and 28S+COI concatenated trees that include relatives with triaenes such as Ancorina robusta (Carter, 1883) and Stelletta fibrosa. Additionally, Stelletta kuhapa sp. nov. shows different haplotypes, but with the current sequence length, it is unclear if these represent different species. There may be intraspecific variation within this species, but a longer sequence fragment would be needed to confirm this diversity. Both Stelletta hokuwanawana sp. nov. and Stelletta apapaola sp. nov. fall within a clade that follows the true definition of Stelletta (Fig. 14) (Cárdenas, 2020). Stelletta hokuwanawana sp. nov. and Stelletta apapaola sp. nov. are 92.5% identical to each other, lack triaenes, and are distantly related (<95% identical) to the other new congeners baring triaenes. However, Stelletta dorsigera Schmidt, 1864, which bares triaenes, is the closest relative to Stelletta apapaola sp. nov. (95.1%). These results show molecular support for the presence/absence of triaenes among sympatric Stelletta spp. but not within the greater context of Tetractinellida diversity.

Figure 14 Maximum-likelihood phylogeny using 28S and 28S+COI partial sequences.

(A) 28S partial sequences from 32 species. (B) 28S+COI partial sequences from 21 species. GenBank acc. no. are included next to the species name in the 28S tree (A) and for the 28S+COI tree (B), acc. no. are given in Table S4. Bootstrap support values >50 are presented at the nodes (2,000 replicates). Species in blue represent the species described in this study. Outgroup used, Cinachyrella apion (28S: HM592753.1, COI: HM592667.1). COI was only successfully obtained for Stelletta kela sp. nov., Stelletta hokunalohia sp. nov., Stelletta kuhapa sp. nov., and Geodia cf. papyracea allowing for a concatenated 28S+COI analysis in B.

Phylogenetic placement of Stryphnus huna sp. nov., based on 28S rRNA sequences showed a close relationship with Stryphnus congeners. All Stryphnus spp. are closely related (95–99% identical) and are strongly supported as a monophyletic clade (bootstrap of 99) in both 28S and 28S+COI trees which shows that the 28S and COI markers might be too conserved to distinguish differences between Stryphnus species. For example, Stryphnus ponderosus (Bowerbank, 1866) and Stryphnus fortis (Vosmaer, 1885) are 99.8% identical even though Stryphnus ponderosus has smaller oxyasters, is found 0–200 m, and lives in the North-East Atlantic and Mediterranean Sea while Stryphnus fortis is a deep dwelling sponges (200–2,600 m depth) and is found in Arctic regions and the Azores. Stryphnus huna sp. nov. is most closely related to Stryphnus radiocrusta (97.9% identity). A defining character for most Stryphnus congeners except for one species (Stryphnus radiocrusta) is the presence of triaenes which Stryphnus huna sp. nov. lacks. The opposite is true for Asteropus sp. congeners where all species are missing triaenes. At this point, there are no other 28S rRNA sequences available for Asteropus sp. which makes it difficult to determine whether Asteropus is a junior synonym of Stryphnus as suggested by Cárdenas et al. (2011). In the meantime, our data adds another example of a Stryphnus sp. which lacks triaenes and is supported by 28S rRNA sequences as close relatives to other Stryphnus congeners.

In 28S and 28S+COI trees, Geodia cf. papyracea from Hawai‘i is closely related (99%) to the G. papyracea specimen from the Caribbean. The 28S tree (Fig. 14A) shows 100% genetic similarity with the Caribbean specimen, while the 28S+COI tree (Fig. 14B) indicates genetic differences. This variation is due to the 213 bp sequence length used for the 28S tree, which excludes two bp. differences expected in longer sequences. A version of the 28S tree that includes these differences in provided in Fig. S1, along with a COI-only tree for additional comparison. A barcoding gap analysis (Meyer & Paulay, 2005) was performed on the 28S+COI tree and revealed a 7% genetic divergence among the Geodia species. This shows that there are sufficient genetic differences across Geodia species. Despite similarities in DNA sequences, more replicates need to be analyzed to confirm intraspecific variation in morphological characters.

Discussion

Cryptic reef environments in Kāne‘ohe Bay exhibit a high diversity of introduced (Vicente et al., 2020; Bettcher et al., 2024) and potentially native or endemic sponges (Vicente et al., 2022a), particularly of species belonging to the class Demospongiae. Here, we integrate morphological characters with phylogenetic placement based on publicly available 28S and COI sequences to describe and determine the geographic origin of seven new records within the second most speciose order of demosponges, Tetractinellida.

Importance of paratype sampling and integrative taxonomy

Integrating molecular and morphological approaches to taxonomy highlights the significance of paratype sampling, revealing a wide range of intraspecific morphological variation with Tetractinellida species, while also demonstrating that classification based on molecular barcodes is concordant with groupings based on spicule composition. Spicule composition is one of the primary taxonomic characters used to distinguish many sponge species and is particularly useful among tetractinellids (Cárdenas et al., 2011). We found that spicule composition was congruent with DNA sequences between conspecific paratypes despite the range of color morphologies within each species. In contrast, external color morphology among Stelletta kela sp. nov., Stelletta hokunalohia sp. nov., and Stelletta kuhapa sp. nov. was a dubious character when differentiating species. For example, regardless of external morphological differences in color, all paratypes within Stelletta kela sp. nov., Stelletta hokunalohia sp. nov., and Stelletta kuhapa sp. nov. were confirmed as conspecifics based on 99–100% 28S rRNA and COI sequence identity (Fig. S1). Molecular markers also improve the resolution of species in cases where external morphologies exhibit noticeable similarities. For example, Stelletta hokuwanawana sp. nov., Stelletta apapaola sp. nov., and Stryphnus huna sp. nov. are all thinly encrusting and share similar color variations of dark grey, white, or grey-brown that make them difficult to distinguish visually. In these instances, the addition of 28S and COI markers detected deep genetic divergence and provided strong support for heterospecificity among these morphologically similar species. The vast differences in morphology between the species described in this study show the importance of integrating molecular phylogeny with traditional taxonomy for the accurate and robust classification of species.

Phylogenetic placement of new Stelletta spp.

Incorporating molecular phylogeny also provides information on the placement of these new Stelletta spp. These new species are placed in two different clades among the tree, splitting up the basal lineage containing Stelletta kela sp. nov., Stelletta hokunalohia sp. nov., and Stelletta kuhapa sp. nov. from the lineage hosting Stelletta hokuwanawana sp. nov., Stelletta apapaola sp. nov. This split supports the presence of triaenes in Stelletta kela sp. nov., Stelletta hokunalohia sp. nov., and Stelletta kuhapa sp. nov. and the absence thereof in Stelletta hokuwanawana sp. nov., and Stelletta apapaola sp. nov. However, Stelletta hokuwanawana sp. nov. and Stelletta apapaola sp. nov. are close relatives to other species that have triaenes present (i.e., Stelletta dorsigera). With the polyphyletic nature of Stelletta, species relationships can be difficult to determine. Both genetic and morphological analyses separate species with triaenes from those without, emphasizing the need for a taxonomic revision of Stelletta.

In previous studies, the synonymy of genera Asteropus and Stryphnus has been proposed due to molecular data showing >95% species identity between genera (Cárdenas et al., 2011). Our data provisionally support this synonymy and we add another example of a Stryphnus species that lacks triaenes. Although, Asteropus spp. currently lacks molecular data to evaluate this synonymy further.

The difficulties of obtaining informative molecular data has certainly affected the pace of new species detections with molecular applications. Limitations in the success of COI barcoding with the standard dgLCO1490 and dgHCO2198 have hindered progress in distinguishing taxa in some families within Tetractinellida, Ancorinidae (<40% success rate) being one of them (Timmers et al., 2020; Vargas et al., 2012). While COI barcodes are generally lacking for Ancorinidae in public databases, 28S markers have been more successful but can evolve slowly in some groups which can fail to distinguish differences between species or even genera. For example, species between Asteropus and Stryphnus can show 95% sequence similarity (Stryphnus fortis and Stryphnus huna sp. nov. share 95% sequence identity), but can also be 99% identical between species (Stryphnus mucronatus (Schmidt, 1868) and Stryphnus radiocrusta share 99.5% sequence identity) (Fig. 14). This marker is not informative for all species comparisons between these two genera which differ substantially in morphological characters.

The potential introduction of Geodia cf. papyracea in Hawai‘i

Geodia cf. papyracea from Hawai‘i exhibited similar morphological traits and 99% sequence identity in nuclear and mitochondrial sequence fragments with a Geodia papyracea specimen from the Caribbean (Cárdenas et al., 2009). This is possibly the first record of G. papyracea in the Pacific Ocean. Geodia papyracea has been consistently and exclusively found across the Caribbean and the Brazilian Atlantic regions (Alcolado, 2002; Bettcher et al., 2023; Burton, 1940; Cárdenas et al., 2009; Cedro et al., 2007; Díaz, 2005; Hechtel, 1965; Muricy et al., 2011; Rützler, 1988; Rützler et al., 2000; Silva, Mothes & Lyrio-Oliveira, 2004; van Soest, 1981a; Wintermann-Kilian & Kilian, 1984) confirming its native distribution and confinement to this region (Fig. 15). In this study, we provide genetic information on the Hawaiian specimen with the possibility of a new record of G. papyracea in the Pacific. Even though, the Hawaiian specimen is 99% identical to the Caribbean specimen, morphological characters differ slightly. These differences in spicule size could be due to environmental parameters. Currently, the range of morphological characters in the Caribbean is unknown. Until there is more data on the intraspecific variation of G. papyracea, we cannot confirm the Hawaiian specimen as a conspecific of this species. Further studies on cryptic communities are essential to learn more about the origin of cryptic sponges and to determine whether they are native, endemic, introduced, or invasive.

Figure 15 Current distribution of Geodia papyracea.

Purple dots represent the specimen from Hawai‘i and the specimen from the Caribbean that is the closest relative (UMPCW921, Cárdenas et al. (2009)). Black dots represent the rest of the distribution (from individual specimens and checklists) throughout the Caribbean and Eastern Brazil. Marine Ecoregions of the World are represented by light grey lines. This map was generated using ArcGIS.

Previously only twelve tetractinellid species were known from Hawai‘i, and all were described over 60 years ago (von Lendenfeld, 1910; de Laubenfels, 1950, 1951, 1954, 1957). Nine of these previous tetractinellid records were dredged at 50–200 m deep while only three (Stelletta debilis, Geodia gibberella, and Jaspis digonoxea) were found at 2–8 m growing on dead coral. Only one of these shallow-water species was known from Kāne‘ohe Bay (J. digonoxea) while the others were recorded from Hawai‘i Island (Stelletta debilis, G. giberella). With this study, the number of known tetractinellids grows from 12 to 19 by surveying a single patch reef location over a span of 2 years. None of the previously recorded species were encountered in our study which not only shows the inconspicuous nature of this sponge group, but also implies that additional species remain to be discovered.

Determining the geographic origin of new Stelletta and Stryphnus species

Although ARMS provides habitat for cryptic communities, they are artificial substrates, and a strict association with artificial or altered environments is proposed by Chapman & Carlton (1991) as a way to recognize alien species. Contrary to the previously recorded tetractinellids, most of the new species were found strictly on artificial substrates which makes it difficult to determine their origin. This strict affinity to artificial habitats could be a reason why they were not recorded in previous studies (von Lendenfeld, 1910; de Laubenfels, 1950, 1951, 1954, 1957), or it could be that the species described here are rare outside of Kāne‘ohe Bay. Among the tetractinellid spp. described here, only Stelletta hokunalohia sp. nov. was found both on natural and artificial substrates, but its distribution and origin remain unknown. Addressing questions of distribution and origin for these sponges will require further studies using ARMS that sample cryptic communities in many more sites throughout Hawai‘i and worldwide.

Conclusion

Using an integrative taxonomic approach has allowed us to describe six new species of tetractinellids from the Hawaiian Islands and one new record potentially introduced from the Caribbean. We provide molecular data from 28S and COI barcoding that show relationships between the Stelletta and Stryphnus species and previously known records, confirming that these are new species. These barcodes also aid in future eDNA studies, which tie these new species with detailed descriptions and molecular data. The use of ARMS allows us to expand our knowledge of Hawaiian cryptic reef biodiversity. Our database of tetractinellid sponges will aid in future identifications around the tropical Pacific as well as provide information on the role these species play in healthy ecosystems. Additionally, these surveys are providing resources to managers for detecting new and introduced species in the field. More taxonomic studies on cryptic sponges can determine the geographical distribution and origination of these cryptic but vital members of the reef community.

Supplemental Information

Supplemental Information 1 Maximum-likelihood phylogeny using 28S and COI partial sequences.

A, 28S partial sequences from 31 species; B, COI partial sequences from 21 species. This includes sequences generated in this study (blue) and sequences downloaded from GenBank. All GenBank accession numbers are included next to the species names. The 28S and COI alignments consisted of 676 bp and 568 bp respectively. Bootstrap support values >50 are presented at the nodes (2,000 replicates). COI was only successfully obtained for Stelletta kela sp. nov., Stelletta hokunalohia sp. nov., Stelletta kuhapa sp. nov., and Geodia cf. papyracea. Outgroup used,Cinachyrella apion(28S: HM592753.1, COI: HM592667.1).

Supplemental Information 2 Museum voucher and GenBank accession numbers for all new species and Geodia cf. papyracea specimens.

FMNH refers to the Florida Museum of Natural History and BPBM to the Bernice Pauahi Bishop Museum. * indicates holotype. – indicates no sequence was obtained.

Supplemental Information 3 Summary of morphological data of known tetractinellid sponges in Hawai‘i.

Supplemental Information 4 Summary of morphological data of known Stelletta, Asteropus, Jaspis, and Stryphnus spp. that share similar morphological characters to the new species described in this study.

*indicates plagiotriaenes and orthotriaenes were described as “ortho- plagiotriaenes” with no distinct separation.

Supplemental Information 5 Species from Figure 14 with COI and 28S GenBank accession numbers.

*indicates outgroup used.

Supplemental Information 6 28S Alignment from Figure 15A.

Supplemental Information 7 COI Alignment from Figure S1B.

Tina Carvalho and Miyoko Belinger are thanked for their assistance and expertise in Scanning Electron Microscopy and histology respectively. We thank Gustav Paulay and Amanda Bemis at the Florida Museum of Natural History and Holly Bolick and Kiana Lee at the Bernice Pauahi Bishop Museum for helping us voucher museum specimens. Paco Cárdenas is thanked for guidance on the initial assessment of the Tetractinellida spp. in this study. These are HIMB 1985 and SOEST 11894 contribution numbers.

Additional Information and Declarations

Competing Interests

Rob J. Toonen is an Academic Editor for PeerJ. The authors declare that they have no competing interests.

Author Contributions

Rachel M. Nunley conceived and designed the experiments, performed the experiments, analyzed the data, prepared figures and/or tables, authored or reviewed drafts of the article, and approved the final draft.

Emily C. Rutkowski performed the experiments, authored or reviewed drafts of the article, and approved the final draft.

Robert J. Toonen conceived and designed the experiments, authored or reviewed drafts of the article, and approved the final draft.

Jan Vicente conceived and designed the experiments, performed the experiments, analyzed the data, prepared figures and/or tables, authored or reviewed drafts of the article, and approved the final draft.

Field Study Permissions

The following information was supplied relating to field study approvals (i.e., approving body and any reference numbers):

Samples from Kāne‘ohe Bay were collected under special activities collection permits SAP2018–03 and SAP2019–06 (covering the period of January 13, 2017, through April 10, 2019) as well as HIMB collection permits SAP2022-22 and SAP2023-31. Samples from 2016 were collected from mesocosms where no permit was required.

DNA Deposition

The following information was supplied regarding the deposition of DNA sequences:

The sequences for the species described are available at GenBank: PQ282243–PQ282295, PQ305253–PQ305258.

Data Availability

The following information was supplied regarding data availability:

The sequences for the species described are available at GenBank: PQ282243–PQ282295, PQ305253–PQ305258.

New Species Registration

The following information was supplied regarding the registration of a newly described species:

Publication LSID: urn:lsid:zoobank.org:pub:F858D9D7-986F-4E56-9C27-B20AC2C12D81.

Stelletta apapaola sp. nov. LSIDL urn:lsid:zoobank.org:act:D203AB34-AA4C-4DFA-AB6D-1610A4FE4284.

Stelletta hokunalohia sp. nov. LSID: urn:lsid:zoobank.org:act:4A8A8BBC-12FE-41DB-8543-FD1CD6D35AFC.

Stelletta hokuwanawana sp. nov. LSID: urn:lsid:zoobank.org:act:1EF50A29-2EE8-491E-B783-5A3A5D7666A7.

Stelletta kela sp. nov. LSID: urn:lsid:zoobank.org:act:D6825A92-3012-4C8C-AAE5-C510FBA1BF73.

Stelletta kuhapa sp. nov. LSID: urn:lsid:zoobank.org:act:776D40C2-33A9-41D5-AD89-108C75362C42.

Stryphnus huna sp. nov. urn:lsid:zoobank.org:act:91AB648C-8712-439D-94AF-26E1689F9657.

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
