# Peer review of "Potential transoceanic dispersal of Geodia cf. papyracea and six new tetractinellid sponge species descriptions within the Hawaiian reef cryptofauna"

_PeerJ, doi:10.7717/peerj.18903_

## Round 0.1 · original submission · Minor Revisions

Dear Dr. Vicente and coauthors,

Thanks for submitting your manuscript to PeerJ. As stated by the two reviewers, this paper makes a good contribution to the description of sponge diversity from Hawaii. Nonetheless, the reviewer (especially reviewer no 1) had few important comments regarding the validity of Geodia papyracea identification, the assignment of taonomic names, and the use of the phylogenetic analysis in the Discussion.

Regarding the last point, I agree with reviewer 1 that this is the weakest point of the manuscript. Although the analyses includes two genes, only 28S is presented as a separate tree, while COI is presented together with 28S in a concatenated tree, potentially masked by 28S. It would be informative to show both gene trees.

Moreover, a barcoding gap analysis could facilitate the species delineation. See for example:

Núñez Pons, L., Calcinai, B. and Gates, R.D., 2017. Who’s there?–First morphological and DNA barcoding catalogue of the shallow Hawai’ian sponge fauna. PLoS One, 12(12), p.e0189357.

Reviewer 1 ·

Basic reporting

This taxonomy paper describes new sponge tetractinellid material from Hawaii, a poorly investigated region in terms of sponge fauna. This is an important first step to start barcoding species from this biodiverse region, which will open up future possibilities to more easily compare sponge species from other parts of the Pacific. Descriptions are overall of very good quality, well-illustrated and completed with COI/28S molecular barcodes. This study would definitely deserve to be published, as a strong addition to the knowledge of Hawaii sponge fauna. There are however several concerns that need to be addressed before this manuscript can be accepted. These will be explained below, followed by additional minor remarks.

I have the following concerns:

1) My main issue is that I still have some doubts about the identity of Geodia papyracea in Hawaii. This is a strong claim and therefore would need to be more strongly backed up. The current evidence is still somewhat weak so I would invite either the authors to be more cautious about their results maybe by calling their specimens Geodia cf. papyracea for now. Alternatively, they could try to gather more data about this: examine comparative material such as type material for example to really compare with their specimens, or try to find and sequence more G. papyracea from the Caribbean. G. papyracea has previously only been found in mangroves, it usually has a parchment-like and brittle cortex, with off-white to light grayish color, with occasional purple tinges, when alive. The sterrasters are significantly larger in G. papyracea (52-75 µm) than in the Hawaii material. You also show several other differences (lines 810-815). All of this in the end makes quite a lot of differences with your specimens. There is clearly some closeness in terms of morphology, and molecular markers, at least to the specimen described in Panama (same COI and 2 bp difference in 28S) but that is not enough evidence. There are many cases in the literature of different sponge species with identical COI. So two options, tune your conclusions down or get additional data to be more convincing.

2) New species names: Adjusting the gender to the genus is only valid for epithets made from adjectives. In the case of last names, you add -ae if you’re honoring a women, and -i if you’re honoring a man. If you’re honoring many people with the same last name, you have to add -orum (this is for several man or a combination of man and women) or -arum (women only). So your first new species epithet should be kaluhiwaorum. Please apply this rule to the five other new species.

3) I think the authors are confused between ‘diagnosis’ and ‘definition’ of a species. Although there is no rule on how to write a diagnose. Taxonomists usually agree that a diagnosis should be limited to only the characters that distinguish it from the other species. Please read e.g. Borkent (2021). There is actually a plea now to formalize this better (Rheindt et al., 2023) so you should be pro-active about it and write proper diagnosis for every new species.
• Borkent A. (2021) Diagnosing diagnoses – can we improve our taxonomy? ZooKeys, 1071.
• Rheindt F.E., et al. (2023) Tightening the requirements for species diagnoses would help integrate DNA-based descriptions in taxonomic practice. PLOS Biology, 21(8), e3002251.

4) In your taxonomic discussions for every species, you never refer to the phylogeny or the sequences. Why is that? These are meaningful characters that can help to separate species and confirm their genus attributions. In my opinion, referring to the phylogenetic tree along your taxonomic results makes more sense. Even more relevant is when you have sequenced several specimens and you find several haplotypes (e.g .for S. camvela), this should be discussed with the taxonomy.

Experimental design

Good quality

Validity of the findings

Good

Additional comments

More minor remarks:

TITLE.

- I would be more cautious about the identification of Geodia papyracea. So I would maybe add the word “potential transoceanic dispersal”.
- I think the word ‘sponge’ is missing in the title. This is obvious for a specialist but not for a non-sponge biologist.

ABSTRACT & KEYWORDS

- Modify the Abstract to be more cautious about G. papyracea identification.
- Keywords: Add ‘Porifera’ as the first key word.

INTRODUCTION

Line 111-112. I would move this sentence to the beginning of paragraph lines 127-141.
Line 112-113. General statement with no references, and not relevant for this study. Please remove.
Line 113-114. Is this specific to tetractinellids? Otherwise, remove this statement, it’s too general.
Lines 115-118. Again, facts about tetracts which are not really relevant for this study. Please remove.
Line 118-125. This part seems to me more connected to the paragraph 143-153. You should probably fuse them.
Line 143-144. Very important sentences in the intro. I would develop it more. Please list all the currently known species of tetracts identified in Hawaii. Please note and explain here to the reader that Myriastra debilis from Laubenfels (1951) is most probably not conspecific with the species described by Thiele (1900) in Indonesia. A simple comparison of the spicules shows this. Also, you forgot to cite Lendenfeld (1910) here, he describes several Erylus from Hawaii (same for line 950).
Line 148-153. Is this text really relevant for this study here? You do not describe any lithistids.

MATERIAL and METHODS

Lines 168-173. Here you should indicate the dates of the sampling, and coordinates for the main sites.
Lines 175-179. What about the spicule preparations, section slides, SEM stubs? Where are they deposited? When specimens have two museum numbers (Table S1) what does this mean? Were specimen split in half? You need to explain this.
Line 185-193. Verb missing in this sentence. Please revise.
Line 187. ‘amplification’ should be with a lower-case letter.
Line 203. I would make your COI and 28S alignments available in the Supplementary material. In case, someone wants to re-analyse your data.
Line 208-209. The sentence on bootstrap and mention of the tree should be removed, this should be in the results.
Line 219. What model of microtome did you use, be specific. How did you embed the sections? What is the % of the nitric acid used?
Line 223-226. It is unclear if you have made permanent slides? I really recommend that you do so, especially when describing new species. These are very valuable later in a museum collection to avoid new slides to be made when researchers want to examine this material.
Line 231. Misleading subtitle. I would change it. You have to make it very clear that you actually haven’t used comparative material in your study. Instead you used literature for comparative purposes.
Lines 233-235. Table S2 should be cited before Table S3. Otherwise, swap the tables.

RESULTS

Lines 251-257. Author of taxonomic ranks should not be in parentheses.
Lines 262. Here the Museum UF# should be cited along with the BPBM#. The reader should not have to search for the UF# in the Supp. Mat, this is a primary piece of information. Please do this throughout the manuscript.
Line 264. With so many paratypes for this and other new species, have you checked the spicules for each of these paratypes? (since they are not all mentioned in the Table 1). I guess you have but I need to ask. Same remark for the other new species.
Line 270. See me comments on “diagnosis” versus “definition”. And correct throughout the manuscript.
Line 275. ‘elevated globular’ sounds a bit strange. I would rather say just ‘globular’, this is usually implies that the sponge is not encrusting and thin.
Line 276. ‘sticky’? Sounds strange. ‘hispid’ is enough I think.
Line 287. Explain where you limit the cortex. This is somewhat unclear in your skeleton description.
Line 288. Replace ‘choanosomal’ with ‘cortical’?
Line 291-292. I would say here the overall structure is ‘radial’.
Line 292-293. The acanthostrongylasters are not present in the cortex? Could you double-check please? I would expect them to be on the top surface?
Line 295. Here I would refer to Table 1, in addition to Fig. 2. Same for the following species. When you describe the spicules, refer to the spicule measurement Table.
Line 319-326. Adjusting the gender to the genus is only valid for epithets made from adjectives. In the case of last names, you add -ae if you’re honoring a women, and -i if you’re honoring a man. If you’re honoring many people with the same last name, you have to add -orum (this is for several man or a combination of man and women) or -arum (women only). So your species epithet should be kaluhiwaorum. Please apply this rule to the five other new species.
Line 334. Add ‘worldwide’ after ‘From 150 Stelletta species’. Here you might want to explain that because of the high risk of invasive species in Hawaii, you have decided to compare your species against all species worldwide.
Line 339-341. Authors of species should not always be in brackets. If the species has been transferred to a different genus then the authors should be in brackets, otherwise, no brackets. For example, you should write Stelletta durissima Bergquist, 1965 and Stelletta fibrosa (Schmidt, 1870). To check this you can use the the World Porifera Database. Please review this throughout the whole manuscript, it’s an important convention. Also, after having written a species name in full with authorship (the first time you mention it), usually you abbreviate the genus. Stelletta durissima becomes S. durissima. Revise this as well throughout the manuscript.
Line 343. You say S. durissima lacks orthotriaenes but orthotriaene measurements are given in Table S3 for that species. Please revise.
Line 348. ‘sclerites’ is not a term used for sponges. What do you mean?
Line 351. Today ‘chiaster’ is not used as much as ‘strongylaster’, which you already use. Please be consistent throughout the paper by using ‘strongylaster’ instead of ‘chiaster’.
Line 352. You should compare with the original description of S. purpurea by Ridley (1884), not a subsequent record (they are more likely to be wrong). Please include the original description in Table S3.
Line 355. You say ‘eight’ species in your intro, not ‘seven’. Please revise. Same remark for line 647.
Line 425. I would remove this first sentence because it’s not entirely correct. It’s usually these megascleres and euasters that suggest Ancorinidae, while here the euasters are probably secondarily lost.
Line 501. According to Supp. Fig S1 you have different haplotypes of 28S in S. camvela. You have sequenced 13 specimens (according to Table S1), so how many haplotypes did you get? Please explain and discuss. This is very unusual, could there be even more species diversity in this species? Is it something you see in the spicule variation? In this case, the judicious designation of the holotype is very important in case you later have cryptic subspecies amongst the paratype material.
Line 507. Space missing in ‘41µm’.
Line 509. I think you mean ‘Western Africa’ not ‘Eastern’ since S. paucistellata comes from Senegal.
Line 579. I think you need to check all species of Jaspis, not only the ones from Hawaii. It is very possible that this species would have been described in the past as a Japis. This remark is also valid for S. brighti.
Line 609. Then I wouldn’t use the word cortex. Maybe it is more of an ectosome (i.e. a few cell layers)? Is there a layer of euasters right at the surface? Your pictures of sections (Fig. 10A-C) are difficult to follow. Could you orient them by designating the surface?
Line 674. Are the oxeas I and II placed in different parts?
Line 675. Usually sanidasters are more abundant at the surface, is that the case?
Line 675 and Line 684. There shoudn’t be any streptasters in Stryphnus species. These are probably just oxyasters with actines growing a bit weirdly (difficult to say anything from your picture 11D). So remove that term from the description, it will confuse readers. Could you try to get better SEM pictures of the oxyasters (I understand that they are rare)? Can you tell if they are spined at least? This would be an important character.
Line 736. ‘Comparative material’ is material you have seen yourself. Is that the case? If not, I would remove this.
Line 752-753. First, this sentence needs to be re-written, I think parts of it are missing. Second, how do you know these are reproductive cells? Ovocytes or spermatocysts? In Fig. 13A, they look more like ovocytes, but please you need to show a better resolution picture of these. In the legend you call them embryos but they are probably not. As far as we know, Geodia species are oviparous so there are no embryos in the adults. For illustrations of both check out the paper:
Koutsouveli, Cárdenas, Conejero, Rapp and Riesgo (2020) Reproductive Biology of Geodia Species (Porifera, Tetractinellida) From Boreo-Arctic North-Atlantic Deep-Sea Sponge Grounds', Frontiers in Marine Science, vol. 7, no. 1091. https://www.frontiersin.org/article/10.3389/fmars.2020.595267.
Line 758-759. Normally, in this species, many plagiotriaenes cross the endocortex and find themselves in the ectocortex at the surface, so that they do not support the cortex, as you write. I think I see some of this in your sections but it’s difficult to tell. Please look at this more closely.
Line 761. Where are the acanthostrongylasters?
Line 770. Saying that there are 3 types of sterrasters is misleading. These are the same category of sterrasters simply at different stages. Usually, one only gives measurements for the mature sterrasters so delete the rest. Likewise, the pictures of immature sterrasters is not informative, you can remove Fig. 13E-F and N. On the other hand, you could make the plate of the spicule much bigger so that the details of the euasters (J and K) can be more visible, here they definitely too small.
Line 791. Replace ‘surfaces’ with ‘pores’.
Line 808. Replace ‘smaller exctocortex’ with ‘thinner ectocortex’.
Line 828-837. Please refer to your tree figures here to help the reader. Here you should mention that this ‘Ancorinidae’ clade is nested within the Geodia clade and is hypothesized to actually be Geodias that have secondarily lost their sterrasters (Cárdenas et al., 2011; Cárdenas, 2020). On the other hand, the two other new Stelletta species group with the Stelletta sensu stricto, those trully belonging to the Ancorinidae. Or you can decide to discuss that information in the lines 898-909.
Line 827. Here you should mention that COI is strictly identical while 28S has a 2 bp. difference. This speaks more to some readers than % similarity. Revise text following my comment to be more cautious about this identification.
Line 861. Spelling: ‘Suberites’.

FIGURES.

I find that overall your pictures of sections are quite dark. Could you lighten them up using photoshop or any other photo-editing software?

Supp. Fig. 1
- Spelling ‘Stryphnus radiocrusta’
- On this tree there seems to be no bp differences within G. papyracea, but there should be 2 bp. difference with 28S. How come?
Table S3.
I would add the measurements of the holotype of Stelletta purpurea.

Reviewer 2 ·

Basic reporting

The paper is clear overall and, for the most part, well-written.
However, there are several places, mainly in the introduction, where the reader would benefit from proofreading by a professional.
please see the specific remarks in the attached PDF
Some remarks:
1. There are some cases in which very high language is used unnecessarily, and other sentences that are poorly written or have grammatical errors.
2. The figure caption will benefit from some elaboration, as the current description is overly concise and technical. Providing additional context or clarifying the key points will help make the figure more accessible.
3. In some places references are missing, I marked a few, but there are more, so please have another look.
4. I believe that Figure S1 should be included in the main article and not as a supplementary figure.

Experimental design

In the methods - the Phylogenetic analysis section needs some elaboration regarding the alignment (see comment in the PDF)

Validity of the findings

no comment

Additional comments

I'm not an expert in the taxonomy of Tetractinellida, however, it does seem that you have done a good job with the taxonomic description.

Annotated reviews are not available for download in order to protect the identity of reviewers who chose to remain anonymous.

---

## Round 0.2 · accepted · Accept

The revision addressed all comments of the reviewers and myself. Please make sure that in the proofs cf. is not italicized. Well done and I am looking forward to seeing this paper published.

Reviewer 1 ·

Basic reporting

I am happy with the responses of the authors, regarding my comments and suggestions.

One important note is that 'cf' in Geodia cf. papyracea should not be written in italics because it is not part of the scientific name. Please correct throughout the manuscript (title, abstract, main text and figure legends).

Scientific names based on personal names do have to follow Latin grammar, according to the current international rules (ICZN). Sorry to hear that the indigenous community did not want to have their names modified. According to the current ICZN Appendix A (Code of ethics), point 4: "No author should propose a name that, to his or her knowledge or reasonable belief, would be likely to give offence on any grounds.". Your new names made from the Hawaiian language describing a character of the sponges are very adequate while honoring the Hawaiian language and culture. Well done.
https://code.iczn.org/appendices/appendix-a-code-of-ethics?frame=1

Experimental design

no comments

Validity of the findings

no comments

Additional comments

no comments

Reviewer 2 ·

Basic reporting

no comments

Experimental design

no comment

Validity of the findings

no comment

Additional comments

The paper is much clearer now.
All of my notes were taken into account and corrected.
Please note that there are a few extra spaces probably because of the edits.